# Insights into Thunderstorm Characteristics from Geostationary Lightning Jump and Dive Observations

Felix Erdmann[1] and Dieter R. Poelman[1]

[1]Royal Meteorological Institute of Belgium (RMIB), Av. Circulaire 3, 1180 Uccle, Belgium

**Correspondence:** Felix Erdmann (felix.erdmann@meteo.be)

**Abstract.** The objective of this study is to gain a deeper understanding of geostationary (GEO) satellite data, with a specific emphasis on sudden increases in a storm's lightning activity, referred to as lightning jumps (LJ), and decreases, known as lightning dives (LD). To achieve this, observations from the Geostationary Lightning Mapper (GLM) and the Advanced Baseline Imager (ABI) on the GOES satellite are utilized to analyze the cloud characteristics of thunderstorms. Storms are then categorized based on whether they produced GEO LJs, GEO LDs, and/or severe weather. While non-severe thunderstorms have a mean cloud top temperature of 236 K, cloud tops are about 20 K colder for severe storms as well as those producing LJs and LDs. Overshooting tops (OTs) in storms producing LJs, LDs and in severe storms were about 3.4 K, 1.9 K, and 2.6 K colder, respectively, than the cloud cell as a consequence of structured and intense updrafts. On the other hand, OTs are rare and shallow in the non-severe storms, and thunderstorms without LJs and LDs. Accordingly, the convective rain rates (CRRs) of the LJ (23 mm/h), LD (20 mm/h) producing storms and severe storms (20 mm/h) are on average more than 3 times higher than in non-severe thunderstorms and storms without LJs or LDs. Thunderstorms experiencing multiple GEO LJs during their lifecycle feature average cloud top temperatures of 213 K, with an average of 0.5 OTs being 4.8 K colder than the anvil, and a mean CRR exceeding 26.4 mm/h. Therefore, especially those storms with multiple LJs have the highest potential to produce dangereous weather events.

## 1 Introduction

Thunderstorms have the potential to give rise to hazardous weather phenomena like strong winds, large hail, flash floods, and tornadoes. A thunderstorm, as its name implies, is defined as a cloud system that produces lightning and thunder. Hence, lightning observations can be used to locate these deep convective systems (e.g., Ávila et al., 2010).

Each storm has its unique lightning characteristics with specific maxima and minima in the lightning activity during the lifecycle of the storm (e.g., Hayden et al., 2021; Borque et al., 2020; Goodman and MacGorman, 1986). Quantifying the changes in the lightning activity means analyzing the time series of the storm cell's flash rate (FR). Rapid increases in the FR are referred to as lightning jumps (LJs) as coined by Williams et al. (1999) while conversely a sudden decrease in the FR can be called a lightning dive (LD). The National Weather Service (NWS) defines severe weather as conditions involving tornadoes, significant hail (with a diameter of at least 2.54 cm or 1 inch), or winds of at least 93 km/h. LJs could be correlated to hail events (e.g., Ni et al., 2023; Nisi et al., 2020; Wapler, 2017; Farnell et al., 2017; Mikuš Jurković et al., 2015), tornadoes

(e.g., Rudlosky and Fuelberg, 2013; Steiger et al., 2007a, b), severe wind events (e.g., Pandit et al., 2023), and also supercell development (Stough et al., 2017). In addition, Schultz et al. (2017) found that LJs result from an intensification of the mixed-phase updraft that also benefits the severe weather production.

While the concept of LJs is well-documented in the literature, lightning dives have rarely been the subject of investigation. The LD exhibits behavior contrary to that of a LJ, leading to a rapid reduction in the FR as first mentioned by Losego et al. (2022). It is based on the idea that a decrease in lightning activity can precede events such as tornadoes or significant hail (e.g., Pineda et al., 2016). That is the case since the rear flank downdraft (RFD) can be related to tornado development (e.g., Satrio et al., 2021; Mashiko, 2016; Markowski, 2002). Within the RFD, internal momentum surges can temporarily weaken the updraft or alter the hydrometeor content. Weaker updrafts prior to tornadoes are reported in previous studies (e.g., Steiger et al., 2007a; Lemon et al., 1978). Such a weakening of the updraft is correlated with reduced lightning activity, as noted by Deierling and Petersen (2008). Furthermore, downdrafts caused by intense rainfall or hail can interact with the storm's updraft and charging structure. These interactions can temporarily reduce lightning activity, as fewer ice particles collide, which is necessary to sustain strong electric fields through non-inductive charging.

The new generation of geostationary (GEO) satellites carries imagers to map the total (i.e., cloud-to-ground [CG] and inter- and intra-cloud [IC]), lightning activity from space. The Geostationary Lightning Mapper (GLM, Goodman et al., 2013; Mach, 2020) provides coverage over the Americas and adjacent oceans, while the Meteosat Third Generation Lightning Imager (MTG-LI, EUMETSAT, 2021b; Dobber and Grandell, 2014) observes, among others Europe, Africa, and the Atlantic. In addition to the GEO lightning data, the new generation of GEO imagers such as the American Advanced Baseline Imager (ABI, NASA, 2022) and the MTG Flexible Combined Imager (FCI, EUMETSAT, 2021a) has seen improvements as well, featuring higher resolution and additional channels, i.e., wavelengths. ABI and GLM provide useful information for nowcasting thunderstorms (Cintineo et al., 2022; Leinonen et al., 2022; Chinchay, 2023). GLM lightning observations have demonstrated potential in the nowcasting of precipitation (with a determination coefficient of approximately 0.6), with limitations in accurately predicting high-intensity rain rates and accumulations (Bourscheidt and Ramos, 2023). Thiel et al. (2020) discriminates between convective and stratiform precipitation by analyzing GLM flash size and frequency. The findings indicate that the most frequent and smallest GLM flashes are associated with the coldest and highest ABI cloud tops (CTs), as well as with overshooting tops (OTs), i.e., signatures of strong convective updrafts.

Different approaches to automatically detect LJs were optimized through verification of the algorithm against the presence of severe weather (Gatlin and Goodman, 2010; Schultz et al., 2009, 2011, 2016). However, in those studies LJ algorithms were tuned based on ground-based lightning mapping array (LMA) data. Curtis et al. (2018) and Murphy and Said (2020) suggested that LJs found for GLM do not resemble LJs identified with LMAs or low frequency lightning location systems as the former are less correlated to radar observations. Erdmann and Poelman (2023) were among the first to optimize the LJ detection specifically for GLM lightning records in the central and eastern contiguous United States (CONUS) and found that GLM LJs as severe weather predictors reach a critical success index of about 0.5, with leadtimes averaging more than half an hour.

Erdmann and Poelman (2023) focused on the automatic detection of LJs from space for the purpose of nowcasting severe weather. In contrast, this study aims to conduct a comprehensive statistical analysis of thunderstorms, including their electrical activity and associated cloud characteristics as observed from GEO satellites. Specifically, it examines how optical GLM LJs and LDs relate to cloud characteristics commonly associated with severe storms, while accounting for detection challenges from space, such as viewing angle, cloud optical thickness, and light scattering. Thunderstorms are then categorized by the presence of LJs, LDs, and/or severe weather reports. Hence, thunderstorms with and without LJs (LDs, severe weather, respectively) can be compared to identify similarities and differences in the satellite-based cloud characteristics. Some previous studies conducted a similar kind of analysis for the LMA-based LJs. Chronis et al. (2015) found that storms with LJs are more organized, more intense, last longer, and exhibit more consistent lightning activity than storms without LJs. This finding was confirmed by Rigo and Farnell (2022) in particular for storms with multiple LJs. LJs could also be related to heavy precipitation events (e.g., Farnell and Rigo, 2020; Wu et al., 2018). The present study aims to determine whether comparable findings and conclusions emerge when utilizing GLM-based LJs and LDs. The key questions to be answered are (i) What do GLM LJs tell us about the storms structure from a satellite point of view?, (ii) Are GLM LJs useful to assess thunderstorm severity?, (iii) Do GLM LDs provide additional information about the thunderstorms?

Section 2 provides information on the datasets and outlines the data processing steps undertaken to derive the results. This encompasses thunderstorm identification, cloud cell tracking, and the detection of LJs and LDs. The subsequent sections, Section 3 and Section 4, delve into the description and discussion of the obtained results.

## 2 Data and Methods

The EUMETSAT satellite application facility (SAF) for nowcasting (NWC) has developed the central software package for this study (Section 2.1). The main source of data is the Geostationary Operational Environmental Satellites R-Series (GOES-R) 16 (former GOES-East) with its ABI and GLM instruments (Section 2.4). Figure 1 introduces the tools and data sources and their relations to each other. Dark grey data are ingested into the NWCSAF software that identifies cloud cells (red) and their satellite-based characteristics (green). Every cloud cell maintains a record of the FR history, allowing the implementation of the LJ/LD detection algorithm (Section 2.6, yellow). LJs/LDs in combination with the severe weather reports (Section 2.3, blue) are used to categorize the cloud cells (purple). The results reveal the characteristics of the different cloud cell categories. Since this study analyzes only the thunderstorm cells, these are termed thunderstorm (TS) categories.

### 2.1 NWCSAF nowcasting software and the RDT package for cell tracking

This work uses identical datasets and software package as in Erdmann and Poelman (2023). Hence, the software package and study periods are briefly introduced below, with more comprehensive details available in Erdmann and Poelman (2023).

The NWCSAF nowcasting software (EUMETSAT, 2022) is a comprehensive nowcasting tool based on satellite data as the prime source of information. NWCSAF v2018.1 (García-Pereda and coauthors, 2019) is used with implementation of technical changes in common modules and on convection products, along with the incorporation of a GLM data reader. This study ingests

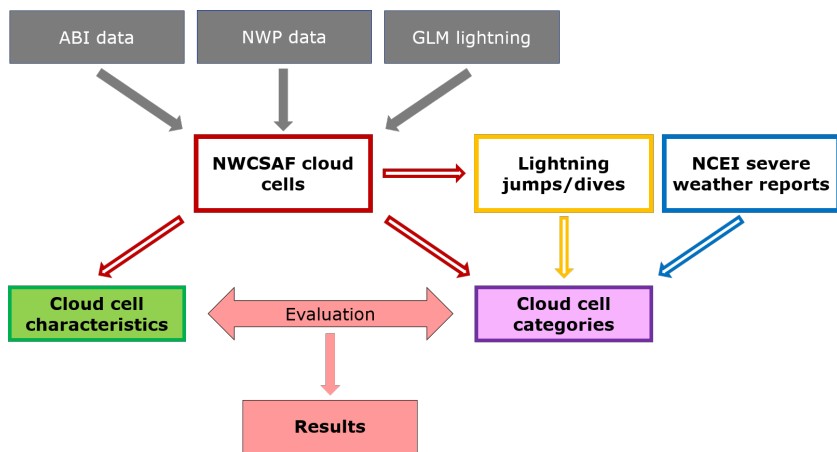

**Figure 1.** Data and product types used in this study. The dependencies of the products are depicted from top to bottom, with arrows also indicating these relationships. At the top, the input is shown in grey. Boxes with colored frames indicate the intermediate products, and the features in colored boxes are analyzed in the Results section.

GOES ABI data (Section 2.4) standard scan with $10\mathrm{minute}$ update cycle as necessary input. To enhance the quality of specific products, especially in cloud cell detection and tracking, data from the European Centre for Medium-Range Weather Forecasts (ECMWF) numerical weather prediction (NWP) and GLM lightning are provided as optional input.

The NWCSAF software is equipped with various modules. The Rapid Developing Thunderstorm Convective Warning (RDT-CW) module (Autones et al., 2020) provides convective cell detection, tracking, and characterisation. The object-oriented approach can effectively differentiate between convective and non-convective cloud cells, and track the convective cells through image recognition, identification of known patterns, and statistical models. The RDT-CW provides outputs for each cell, including the cell contour, various physical cloud characteristics (as detailed in Section 2.5), information about brightness temperatures (BTs) and reflectances, OTs, convective rain rates (CRRs), and the GLM flash rate (FR). Additionally, RDT also
corrects for satellite parallax effects.

    OTs define a region of the cloud top that exceeds the surrounding cloud shield, often seen as a dome above an anvil (e.g., Bedka and Khlopenkov, 2016). OT development needs a strong force manifested as a strong, persistent updraft in thunderstorms. Hence, OTs are indicative of dynamical thunderstorm cells with strong updrafts that are usually well organized. Given
that strong updrafts frequently play a crucial role in the formation of tornadoes and large hail, storms with these characteristics are especially significant for nowcasting. Since OTs are usual transient features, this study analyzes the maximum OT activity of each thunderstorm. OTs are detected in the NWCSAF RDT package through the application of several temperature and brightness temperature difference (BTD) criteria. The software identifies extremely cold cloud pixels (colder than 223 K in the mid-latitudes), and compares them to the surrounding pixels to identify the depth (as the temperature difference, DT) and
horizontal extent of the OT. The BTDs take channels of WV6.2, WV7.3, and IR10.8 into account. Satellite pixels can also be

**Table 1.** Study periods and the number of analyzed thunderstorms (full trajectories) in the CONUS per period (excluding the spin-up time of 3h and instrument downtime).

| Period | Number of storms |
|--------|------------------|
| Jan 10-11, 2020 | 844 |
| Feb 04-06, 2020 | 852 |
| Jun 02-10, 2020 | 11256 |
| Aug 14-16, 2020 | 5414 |
| Nov 24-25, 2020 | 564 |
| Jan 25-16, 2021 | 815 |
| Feb 13-15, 2021 | 352 |
| Apr 08-10, 2021 | 1313 |
| Aug 30-31, 2021 | 3563 |
| Overall | 24973 |

identified as OT if they are at least 5 K colder than the tropopause. A detailed description of the OT detection algorithm can be found in Autones et al. (2020, p. 49-50).

The NWCSAF software also includes a dedicated package to estimate CRRs. This estimation uses analytic functions calibrated on radar data as ground truth, and also takes lightning observations into account. The complex algorithm to estimate CRRs is detailed in Lahuerta et al. (2020, p. 22-41).

## 2.2 Study days

Study days are selected based on the following aspects: (i) There is a spinup for each NWCSAF software run of 3 hours as a trade-off between included data and negative effect on RDT during the beginning of the run. Hence, selected periods of more than 24 consecutive hours are prefered for efficiency. (ii) Each period should contain storms with different severe weather types ensuring a minimum of two among wind, hail, and tornado reports during the period's duration. (iii) The overall dataset should cover different seasons. (iv) GOES ABI and GLM data must be available. It is worth noting that there was one relevant GOES-16 downtime from 03 June 17:00UTC to 04 June 01:30UTC (Table 1).

An RDT cloud cell with matched GLM flashes defines a thunderstorm. This study aims to understand the meaning of LJs and LDs for thunderstorm characteristics. RDT cloud cells without lightning activity are excluded from further study, as they are typically stratiform phenomena, shallow convection, or cells in their early development or dissipation phase.

## 2.3 Severe weather reports

The National Centers for Environmental Information (NCEI) weather database collects reports of human observers to archive the frequency and impact of significant weather events in the U.S. that may cause loss of life, injuries, significant property

damage, and/or disruption to commerce (NCEI-NOAA, 2020). The reports are validated by experts, hence, there is a quality control for the reports within the database. The reported events encompass a variety of types, ranging from severe weather events such as tornadoes, large hail, and thunderstorm winds, to extreme temperatures and rare, unusual weather phenomena. This study uses the severe weather reports indicated as tornado, large hail, and thunderstorm winds for the study periods introduced in Section 2.2.

A density-based clustering algorithm (DBSCAN scikit-learn developers, 2007-2022) groups all reports of the same type (i.e., tornado, hail, wind) that occurred within 10km and 6minutes (Erdmann and Poelman, 2023; Schultz et al., 2016). The cluster of reports that is created is referred to as a severe weather event whereby the time and location of the event correspond to its first report. To allocate the severe weather events to RDT cloud cells, cloud cells are considered at the exact time of a weather event. Therefore, the RDT cells are shifted using their motion vectors. An NCEI event belongs to a cloud cell if it is found within the cloud cell contour at the time of the event. For NCEI events that do not fall inside any cloud cell contour, a distance of 50km around the event is also considered to assign it to the closest RDT cloud cell within that radius. As a result, RDT cloud cells receive an additional attribute indicating whether they produced a tornado, hail, and/or wind report.

## 2.4 ABI and GLM data

ABI on GOES-R satellites observes the Western Hemisphere's weather, oceans and environment. The passive multichannel radiometer has 16 different spectral bands including two visible channels (at 0.5- and 1.0-km resolution), four near-infrared channels (at 1.0-km resolution), and ten infrared channels (at 2-km resolution) with on-orbit calibration. Each channel views specific aspects of the atmosphere or surface such as trees, water, clouds, moisture or smoke (NASA, 2022) providing unique information. Several products can be deduced including cloud top details such as height and phase, storm motion vectors, radiation products, land and sea temperatures, surface type, albedo, aerosol information, and fire and volcanic ash characterization. Applications include the monitoring of cloud formation, tracking severe weather, assessing fire, smoke, and air quality, as well as understanding ocean dynamics.

Only GOES-16's ABI is used here. Although this study analyses the western and central CONUS, where the ABI rapid scan is available, ABI standard scan with updated images every 10minutes is used, with the region limited to the CONUS. This aids in efficiently running the NWCSAF software and reducing the data volume.

GLM features optical detection of the light emitted by lightning, which is visible on the cloud top or edges. It monitors the total lightning activity from GEO orbit with narrow-band sensitivity of 1nm within the 777.4nm oxygen band. The variable pitch pixel charge coupled device (CCD) reduces the effect of increasing pixel size towards the edge of the field of view (FoV). Hence, pixels measure 8km nadir and 14km at the edge of the FoV (Goodman et al., 2013). GLMs wide angle lense covers nearly the full disk (1372×1300 pixels). The primary detected elements are single illuminated pixels, referred to as events. Adjacent events of the same 2ms time frame form a group. Groups are clustered to flashes by a weighted euclidean distance (WED) approach with 16.5km latitude and longitude and 0.33s temporal constraints (Mach, 2020). The impact of the GLM performance and variations of it over the CONUS are discussed in Appendix A. GLM flashes are ingested into the NWCSAF software. RDT then assigns the GLM flashes to the cloud cells, whose position relative to the flash radiance-weighted centroid

is checked at the exact time the GLM flash occured. The software outputs the 1-minute time series of the flash rate (FR) for each cloud cell.

## 2.5 Thunderstorm characteristics and the normalization

In total, this study analyzes 14 thunderstorm characteristics (Table 2) that are deduced from ABI channels directly (i.e., BT and BTD) or provided by the RDT software based on ABI observations (e.g., rain rates and OTs). These characteristics are expected to identify a thunderstorm, and a comparison should be made across different TS categories. To facilitate the comparison and illustration of the results (see Figure 3), the characteristics are normalized following Equation (1). The minimum and maximum values for each characteristic are taken from all analyzed thunderstorms and do not depend on the TS category. Hence, normalized characteristics can still be compared between different categories. The range of 0 to 1 indicates whether a certain characteristic received low or high values for the analyzed category relative to all other thunderstorms.

$$x_n = \frac{x - min(X)}{max(X) - min(X)} \tag{1}$$

with $x_n$ representing the normalized value of a characteristic, ranging from 0 to 1, $x$ is the specific value of the characteristic for TS category being analyzed, $X$ denotes the entire set of values for the characteristic from all TS categories, encompassing all analyzed thunderstorms, and $min(X)$ and $max(X)$ are the minimum and maximum values of this characteristic across the entire set of values, respectively.

## 2.6 Lightning jumps and lightning dives

The LJ algorithm used in this study is the FRarea LJ algorithm, optimized for GLM lightning records as detailed by Erdmann and Poelman (2023). With a FR threshold of 15 flashes per minute and a sigma level of 1.0, the algorithm first checks if the current FR exceeds the given threshold of 15 flashes per minute and only then proceeds with the subsequent steps. The FR time series is smoothened and normalized to obtain a 2-minute averaged. It then divides the FR by the RDT cloud cell area at that specific time to obtain an area-normalized FR. The discrete time derivative of this normalized 2-minute FR, referred to as DFRDT, is calculated. The $\sigma$ value is obtained from the standard deviation of the DFRDT of the previous 5 (i.e., not including the most recent DFRDT) 2-minute time steps. The ratio of the most recent DFRDT to $\sigma$ is called the $\sigma$-level and serves as the LJ detection threshold. If the $\sigma$-level exceeds the given threshold of 1.0, a LJ is detected. LJs that occured within 6 minutes and also newly detected LJs at the time of ongoing LJs are merged to one long-lived LJ (compare Schultz et al., 2009).

LDs are obtained by the same algorithm when using negative $\sigma$-levels. The CSIs of the LD algorithms are initially calculated when verifying NCEI weather events for all analyzed thunderstorms, with the same verification method as for the LJs in Erdmann and Poelman (2023). The applied LD algorithm with highest CSI makes use of the FRarea algorithm with FR threshold of 10 and $\sigma$-level of -1.0.

Figure 2 illustrates the application of the LJ and LD detection algorithms for one thunderstorm trajectory starting on 06 Feb. 2020 at 0520 UTC and lasting almost 90 minutes. The thunderstorm reached a maximum FR of 48 flashes per minute about

**Table 2.** Thunderstorm (TS) characteristics.

| Characteristic | Description [unit] |
|---|---|
| cell area | maximum area of a cell in the trajectory [km2] |
| IR12.3(min_BT) avg | average over minimum BTs in IR12.3 channel for the cells of the trajectory [K] |
| min T avg | minimum of the cell-averaged BTs for the trajectory [K] |
| min pressure (top) | minimum pressure of any CT pixel for trajectory [hPa] |
| vertical grad(T) | average vertical temperature gradient (absolute) of cells in the trajectory [K/km] |
| cloud ice fraction | fraction of pure ice ABI pixels to mixed-phase and liquid water pixels [-] |
| IR3.9(min_BT) avg | average over minimum BTs in IR3.9 channel for the cells of the trajectory [K] |
| overshoot count max | maximum number of OTs for one cell of the trajectory [-] |
| overshoot DT max | maximum IR11.2 BTD between pixels of the OT and the surrounding pixels for cells of the trajectory [K] |
| max CRR | maximum convective rain rate for cells of the trajectory [mm/h] |
| WV6.2(min_BT) avg | average over minimum BTs in WV6.2 channel for the cells of the trajectory (upper level water vapor) [K] |
| WV7.3(min_BT) avg | average over minimum BTs in WV7.3 channel for the cells of the trajectory (mid-level water vapor) [K] |
| WV6.2-WV7.3(p90) max | maximum of the 90th percentiles of WV6.2-WV7.3 BTDs for the cloud cells of the trajectory [K] |
| WV6.2-IR11.2(p90) max | maximum of the 90th percentiles of WV6.2-IR11.2 BTDs for the cloud cells of the trajectory [K] |

75 minutes after the cell had been identified, with a second FR peak observed 54 minutes after the start. In total, 2 LJs and 3 LDs were detected as indicated by the red markers. The detection algorithm thresholds are also shown as horizontal lines, in blue for the FR threshold and in grey for the $\sigma$-level threshold. The flash rate must be greater than the FR threshold to detect a LJ or LD. At the same time, the $\sigma$-level should exceed the threshold for the LJ algorithm, and be more negative than the threshold for the LD algorithm. The $\sigma$-level peaked during the first LJ. Although the raw FR increased more rapidly during the second than during the first LJ, the simultaneous growth of the cell led to a smaller $\sigma$-level than in the first LJ, as the LJ algorithm accounts for cell area by dividing FR by the cloud cell area.

## 2.7   Thunderstorm (TS) categories

Thunderstorms are categorized based on the presence and absence of LJs, LDs, and NCEI severe weather events. Table 3 presents the analyzed TS categories that emerge from this process with the associated number of thunderstorm trajectories in each category.

## 3   Results

From Table 3 it follows that the vast majority of thunderstorms do not produce a LJ (95.9 %), a LD (91.4 %), and/or severe weather (96.1 %). The categories labeled "withTornado", "withHail", and "withWind" include the thunderstorms that produced tornadoes, hail, or wind, respectively. The total count of these categories (79+438+645=1162) exceeds the number of severe

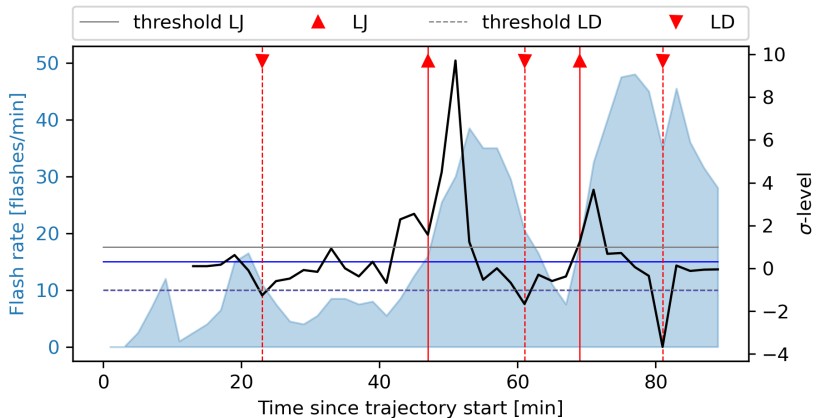

**Figure 2.** Flash rate (blue shading), $\sigma$-level (black) and detected LJs and LDs (red triangles) for one thunderstorm trajectory starting 06 Feb. 2020, 0520 UTC. The FR-thresholds (blue) and $\sigma$-level thresholds (grey) of the LJ (solid) and LD (dashed) detection algorithms are shown as horizontal lines (dashed lines superimposed).

**Table 3.** Thunderstorm (TS) categories and the number (n) of full trajectories in each category.

| TS category (short name) | Number (n) |
|---|---|
| all | 24973 |
| with LJ (withLJ) | 1031 |
| with 1 LJ (singleLJ) | 519 |
| with multiple LJs (multiLJ) | 512 |
| without LJs (noLJ) | 23942 |
| with LD (withLD) | 2136 |
| with 1 LD (singleLD) | 1464 |
| with multiple LDs (multiLD) | 672 |
| without LDs (noLD) | 22837 |
| without LJ and with LD (noLJ & LD) | 1105 |
| with NCEI report, severe (withNCEI) | 970 |
| with tornado report (withTornado) | 79 |
| with severe hail report (withHail) | 438 |
| with severe wind report (withWind) | 645 |
| without NCEI report, non-severe (noNCEI) | 24003 |

thunderstorms (970), indicating that several thunderstorms produced more than one type of severe weather. It is noted that twice as many thunderstorms had LDs than the number of storms with LJs.

**Table 4.** Mean values of the characteristics for selected (TS) categories. The categories *withLJ* (*withLD*) combine singleLJ and multiLJ (singleLD and multiLD), and withNCEI combines withTornado, withHail, withWind.

| Characteristic (mean) | withLJ | singleLJ | multiLJ | noLJ | withLD | withNCEI | noNCEI |
|---|---|---|---|---|---|---|---|
| cell area [km2] | 15,780 | 10,373 | 21,262 | 1,911 | 10,460 | 12,812 | 2,066 |
| IR12.3(min_BT) avg [K] | 213 | 215 | 211 | 235 | 217 | 218 | 234 |
| min T avg [K] | 215 | 217 | 213 | 237 | 218 | 219 | 236 |
| min pressure (top) [hPa] | 128.5 | 136.9 | 120.1 | 221.9 | 138.6 | 152.7 | 220.8 |
| vertical grad(T) [K/km] | 11.8 | 13.9 | 9.6 | 21.8 | 14.4 | 13.2 | 21.7 |
| cloud ice fraction [-] | 0.94 | 0.92 | 0.97 | 0.79 | 0.92 | 0.93 | 0.79 |
| IR3.9(min_BT) avg [K] | 234 | 236 | 232 | 252 | 237 | 241 | 252 |
| overshoot count max [-] | 0.34 | 0.21 | 0.47 | 0.0 | 0.20 | 0.27 | 0.01 |
| overshoot DT max [K] | 3.4 | 2.1 | 4.8 | 0.0 | 1.9 | 2.6 | 0.1 |
| max CRR [mm/h] | 23.2 | 20.1 | 26.4 | 5.8 | 19.5 | 19.5 | 6.0 |
| WV6.2(min_BT) avg [K] | 211 | 213 | 210 | 225 | 214 | 215 | 225 |
| WV7.3(min_BT) avg [K] | 212 | 214 | 211 | 230 | 215 | 216 | 230 |
| WV6.2-WV7.3(p90) max [K] | 0.1 | -0.2 | 0.3 | -3.3 | 0.3 | -0.5 | -3.2 |
| WV6.2-IR11.2(p90) max [K] | -1.2 | -1.8 | -0.6 | -8.0 | -2.0 | -2.3 | -8.0 |

The ABI-based cloud characteristics (Table 2) are analyzed to comprehend the significance of GLM LJs and LDs. The results for the LJ storms are primarily discussed, and LD are included in the discussion where the findings differ compared to the LJs. It should be emphasized again that this paper investigates GEO-based LJs and LDs from optical lightning detection, in contrast to former studies that analyzed ground-based detected VHF or LF LJs.

## 3.1 Characteristics of the TS categories

This section presents key findings on the characteristics of thunderstorms, based on the results in Figure 3 and Table 4. Figure 3 compares normalized characteristics (Section 2.5) among thunderstorms with and without LJs, while Table 4 provides the mean values for each characteristic across selected TS categories. Finally, Figure 4 also shows the distributions of three selected characteristic to compare all TS categories in more detail: (a) cloud ice fraction, (b) maximum CRR, and (c) WV6.2-IR11.2 BTD.

### 3.1.1 Thunderstorm cell area

First, the normalized distributions of the observed cell areas for storms without (Figure 3a) and with LJs (Figure 3b) are compared. Both, the mean (red cross) and even the median (red line) cell area for storms with LJs are higher than the 75th percentile (box) for the noLJ storms. Hence, LJ producing thunderstorms have significantly larger footprint areas than those

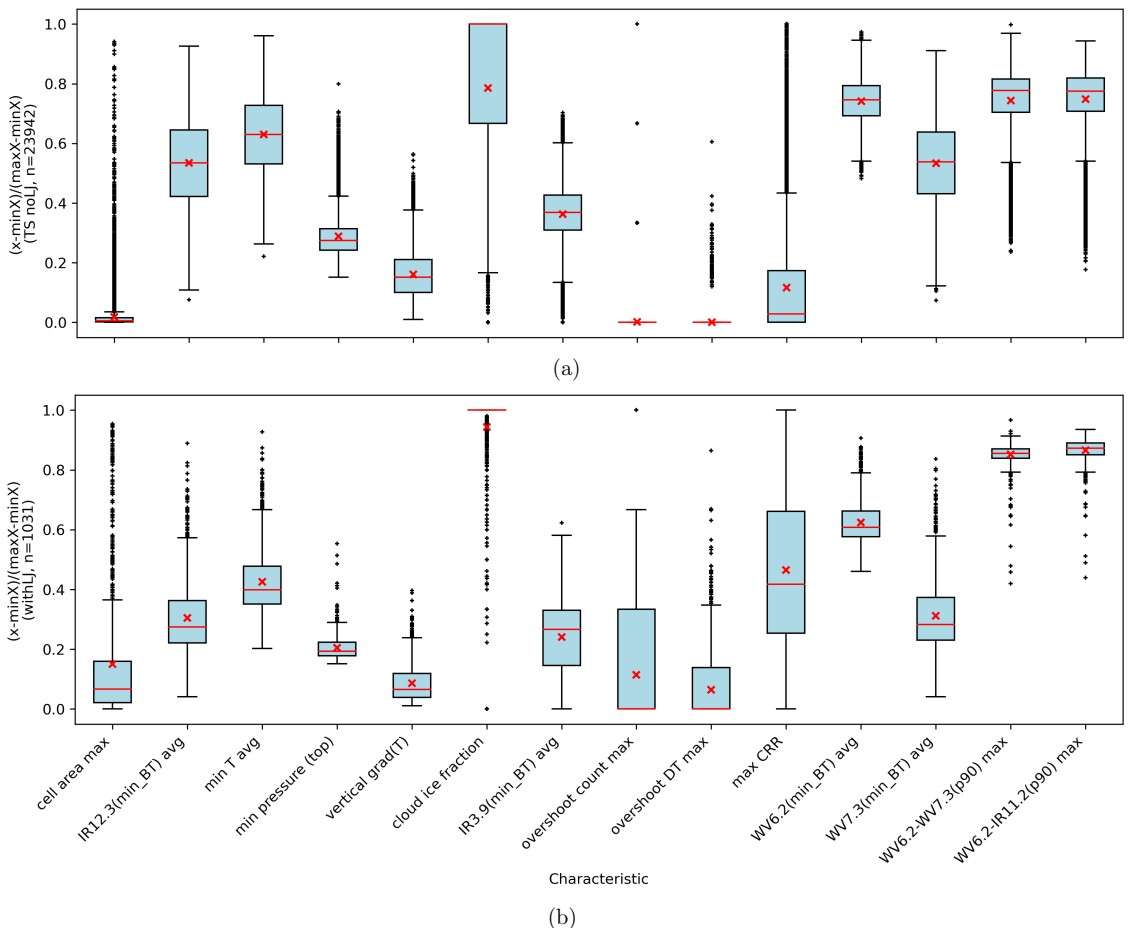

**Figure 3.** Normalized characteristics for (a) the thunderstorms (TSs) without LJs and (b) the storms with LJs (singleLJ + multiLJ). Red cross shows the mean, red line the median of each characteristic.

without LJs. The latter category still contains a few very large storm cells, indicated by the outliers in Figure 3a. In general, those are found in each TS category which suggests that satellite-based cell detection cannot always seperate single cells under a continuous cloud shield.

The mean cell areas in Table 4 confirm the previous finding. On average, severe thunderstorm cells covered an area of 12,812 km2 (median 4,089 km2). Storms with LJs had an average area of 15,780 km2 (median 6,995 km2), while non-severe thunderstorms and those without LJs typically covered about 2,000 km2 on average, with medians around 550 km2. The multiLJ and tornadic storm cells were the largest and covered on average over 20,000 km2 (medians over 9,000 km2). Thunderstorms with LJs cover a larger area than the storms with LDs, and cells of both these categories are larger than the thunderstorms without GLM LJs. This could be related to the formation of large anvils for CTs near the tropopause that acts as a natural ceiling. If a thunderstorm grows up to the tropopause, vertical development is hampered and moist air is forced

horizontally. The satellite-based cell detection sees the resulting anvil and those cells appear larger than thunderstorms that grow mainly vertically. The fact that thunderstorms with LJs, LDs and the severe storms covered larger areas than the average area of all thunderstorms may indicate an above-average fraction of well-organized thunderstorm types like supercells, multi-cell storms, or MCSs that are known for larger footprint areas than the ordinary single cell thunderstorms.

### 3.1.2 Cloud top characteristics and overshooting tops

In general, the storm cells with LJs and/or LDs have colder CTs compared to storms without LJs and LDs. The 75th percentile of CTs in storms with LJs and/or LDs remains colder than the 25th percentile of CTs in storms without these events, as shown by the interquartile ranges (IQR) highlighted in blue in Figure 3. Coldest CT temperature is found for the multiLJ TSs with a mean value of 213 K. That is colder than typical BT thresholds for the detection of overshooting tops in the range of 215 K and high anvils with 225 K (Autones et al., 2020; Bedka and Khlopenkov, 2016). The categories noLJ, noLD, and noNCEI have

CTs warmer than 235 K, warmer than any other TS category. The CT temperatures of withLJ storms match those of severe storms, and storms without LJs have warmer CTs similar to non-severe storms (Table 4). In agreement with these results, the TS categories with the coldest CTs have the lowest CT pressure (average about 110-120 hPa). The strongest vertical temperature gradients occurred within the TS categories noLJ, noLD, noNCEI also featuring the lowest CTs (Table 4) since the vertical temperature gradient decreases towards the tropopause and eventually inverts to increasing temperature with height

in the stratosphere.

The analysis of the CT phase confirms the previous findings and shows that the cloud physics are in accordance with the BT measurements. Cloud ice fraction averages above 0.95 for the TS category multiLJ and 0.94 for LJ storms, thus, most of those cells consist of ice-phase ABI pixels only. The mean cloud ice fraction is below 0.8 for the thunderstorms without LJs, LDs, and/or NCEI events (Table 4). However, the majority of the cloud is glaciated for all thunderstorms as the median of cloud ice

fraction equals 1 for all categories. Figure 4a also demonstrates that severe TSs feature high cloud ice fraction similar to the LJ storms and LD storms. The 3.9 $\mu$m channel is useful to gain some insight into the ice crystal contents. Small ice crystals reflect more of the solar radiation of 3.9 $\mu$m than large crystals. Hence, colder BTs in the IR3.9 channel indicate larger ice crystals within the storms with LJs and those with LDs than for the noLJ storms (Table 4). Large ice crystals, graupel and hail can particularly form in strong convective updrafts where they have time to grow.

If graupel forms within the updraft region, it can then collide with small ice crystals and lead to non-inductive charging, the major cloud electrification process in extratropical thunderstorms (e.g., MacGorman and Rust, 1998). Hence, updraft strengths are well correlated with storm FRs (see also Deierling and Petersen, 2008). An updraft intensification can cause a LJ and is favorable for severe weather, and strong convective updrafts can also cause OTs. Most and strongest OTs occured in thunderstorms of the categories multiLJ with mean count of 0.47 and mean OT DT max of 4.8 K (Table 4). Among the severe

thunderstorms with 0.27 OT count and mean OT DT max of 2.6 K, the withTornado (0.62, 5.3 K) and withHail (0.42, 3.8 K) storms stand out (not shown). Hence, those and the multiLJ storms feature the most persistent and strongest updrafts. The counts of OTs are higher in thunderstorms with LJs and/or LDs compared to the storms without LJs and LDs (Figure 3, Table 4). Mean and even the 75th perentile of OT count max equal 0.0 for thunderstorms without LJs, making OTs rare exceptions

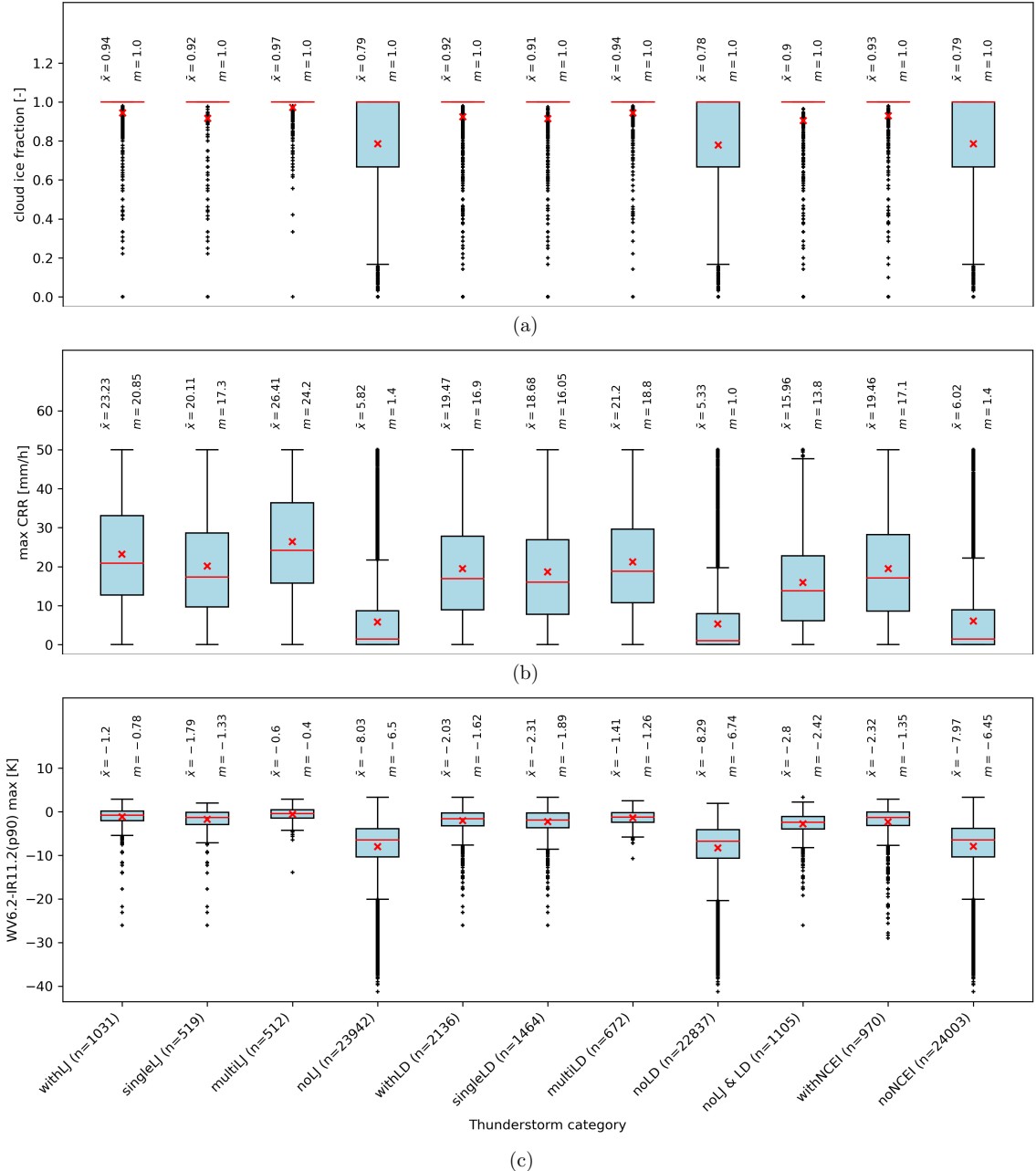

**Figure 4.** Distributions of (a) the fraction of pure ice pixels to mixed-phase and liquid water pixels (cloud ice fraction), (b) maximum estimated CRR during the cell lifecycle, (c) BTDs of WV6.2-IR11.2 as the maximum of the 90th percentiles BTD for each time step during the cloud cell lifecycle for the thunderstorm cell categories. $\bar{x}$ shows the mean, $m$ the median for each category.

for the non-LJ storms (Figure 3a). Hardly any OTs are seen for the thunderstorms without LDs and the non-severe storms, too. The behavior matches the expectation to see more and stronger OTs, i.e., higher OT DT max, in the severe than the non-severe storms (e.g., Bedka, 2011). Morevover, the LJ and LD storms have OT counts (mean of 0.34 and 0.20, respectively) and OT DT max (mean of 3.4 K and 1.9 K) that resemble the patterns observed in severe storms (means of 0.27 and 2.6 K).

### 3.1.3  Rain rates and water vapor

The max CRR reveals that thunderstorms with LJs experience higher rain rates than storms with LDs also seen from Figure 4b, while CRRs of the latter are still significantly higher compared to the noLJ storms (Table 4). Furthermore, the thunderstorms with LJs and those with LDs have lower BTs for both WV6.2 and WV7.3 channels compared to storms without LJs (Figure 3). High water vapor content means high amounts of water being stored in the atmosphere that could be released as precipitation resulting in high CRRs. Both the WV6.2 and the WV7.3 channels exhibit the lowest BTs for the multiLJ thunderstorms. In addition, thunderstorms that produced tornadoes and/or severe hail contain more water vapor in the mid and upper levels than the severe wind storms (not shown).

Detailed statistics on the maximum CRR are presented in Figure 4b. The TS categories noLJ, noLD, and noNCEI consistently show the lowest maximum CRRs, with mean values below approximately 6 mm/h and a median of less than 1.5 mm/h. Additionally, the 75th percentile for these categories remains below the 25th percentile (IQRs, shown in blue boxes) of the other TS categories. The highest averages of thunderstorm max CRR are observed for the category multiLJ (26.4 mm/h). Thunderstorms with LDs have average max CRRs of 19.5 mm/h, thus, somewhat lower than the storms with LJs (23.2 mm/h) but similar compared to all severe storms (19.5 mm/h). Among the severe storms, the category withTornado (mean: 27.6 mm/h) has higher max CRR than severe hail (21.0 mm/h) and wind storms (18.9 mm/h). However, one needs to consider the high variability of CRRs in each TS category expressed as IQRs of about 20 mm/h. These satellite-based CRRs agree well with results of Feldmann et al. (2023) that found radar-derived rain rates in the range of 20 mm/h to 30 mm/h for hailstorms and supercells, thus, organized convection, and rain rates below 10 mm/h for ordinary thunderstorms. The results for the mean and median CRR during thunderstorm lifecycles lead to similar conclusions as those presented for the maximum CRR. Hence, the storms in the stated categories with high CRR produce significant amounts of rainfall throughout their entire lifecycle. Overall, the results for water vapor content align with those for CRR. However, tornadic storms with the highest CRRs across all categories are notable, even though they may not have the highest single water vapor content.

### 3.1.4  Brightness temperature difference

BTDs are commonly used in satellite science since they combine information from different channels. For example, IR11.2 alone gives information about the CT temperature, however, it does not tell anything about the clouds below. Combining IR11.2 and WV6.2 (Figure 4c) provides information about the CT and upper level water vapor content. BTDs as defined in this study (Table 2) have in general negative values for cloud cells. The BTD gets closer to 0 or becomes slightly positive for the deep convective clouds. Hence, the higher the BTD is, the more organized is the convection and the cloud cell. Mean BTDs are significantly higher for the TS categories with LJs, LDs, and/or NCEI reports. Storms with GLM LJs and LDs typically form

in regions characterized by high levels of upper-level moisture and evolve through the intensification of deep convection. For example, the WV6.2-IR11.2(p90) max averages -2 K to -1 K for the categories withLJ, withLD, and withNCEI, and even above -1 K for the multiLJ thunderstorms (Figure 4c). The means for TS categories without LJs, LDs, and NCEI reports are in the range of -9 K to -8 K. Figure 4c illustrates that high negative BTDs below -20 K of WV6.2-IR11.2(p90) max are mainly found for the thunderstorms without LJs, LDs, and NCEI reports. These low BTDs indicate shallow convection. Overall, the BTDs exihibt similar statisitical distributions for storms with GLM LJ and/or LD and for the severe thunderstorms.

## 3.2 LDs

LJ storms generally have slightly colder CT temperatures (215 K vs. 218 K) and lower CT pressures (129 hPa vs. 139 hPa) compared to the LD storms. They also cover a larger area (15,780 km2 vs. 10,460 km2), exhibit higher average CRRs (23.2 mm/h vs. 19.5 mm/h). Additionally, LJ storms produce, on average, more (0.34 vs. 0.20) and stronger OTs (maximum DT of 3.4 K vs. 1.9 K) compared to thunderstorms with LDs (see Table 4). In consequence, the LJ detection has a stronger correlation to the most organized convection with strong updrafts than the LD detection. LDs occured also in storms with weaker updrafts and lower CTs. However, there are also severe weather events that occur in shallow convection and storms with weaker updrafts (i.e., no OTs). There were 188 severe thunderstorms with a LD but no LJ. 38.0 % and 31.6 % of severe and tornadic thunderstorms, respectively, had no LD, compared to 57.4 % and 51.9 %, respectively, for LJs. The relatively high probability to detect tornadic storms with LDs agrees well with the idea of the RFD interacting with the updraft to cause a temporary drop in the FR and also playing an important role in tornadogenesis. It should be noted that the thunderstorms with LDs contain among others all the thunderstorms with LJs, but about half of the LD storms had no LJ.

## 3.3 Single LJ versus multiple LJ storms

The previous sections compared storms with LJs to storms without LJs and to storms with LDs. This section puts emphasis on the differences that are found for characteristics of thunderstorms with multiple and single LJs. The average values (see Table 4) are used since the distributions of the characteristics feature similar shapes for single and multi LJ storms, as seen for the three examples in Figure 4. Multiple LJ storms have slightly colder and higher CTs than single LJ storms with CT temperatures average 213 K and 217 K, respectively. Thunderstorms with multiple LJs during their lifetime manifest the deepest convection. OTs are twice as frequent (0.47 versus 0.21) and significantly stronger (DT max of 4.8 K versus 2.1 K) in storms with multiple LJs compared to those with only a single LJ. The 75th percentile of both the OT count and the OT DT max distribution for storms with only one LJ remains zero. In contrast, the 75th percentiles of these characteristics reach 1.0 and 9.8 K, respectively, for the TS category multiLJ. Strong, organized updrafts occur mostly within the multiLJ storms. However, the water vapor channels and BTDs yield similar values for the multiLJ and singleLJ storms (see Table 4). Both TS categories contain deep convective cells that form in similar environments. Hence, the updraft strength remains a major difference between multiLJ and singleLJ storms. The max CRR of multiLJ storms (26.4 mm/h) clearly exceeds that of singleLJ storms (20.1 mm/h). This implies that the storms with multiple LJs are more prone to experiencing the highest rain rates, posing an elevated risk of flash floods compared to storms with only one LJ (see also Figure 4b for max CRR

distributions). All these findings for GLM-based LJs are consistent with the results for LJs on the flash level data reported by Rigo and Farnell (2022), who analyzed ground-based multi-LJ storms using a different cell tracking method. Specifically, Rigo and Farnell (2022) suggest that convection in multi-LJ storms is more organized compared to other cases, as these storms sustain high radar variable intensity over extended periods.

### 3.4 Summary

For the first time, thunderstorms with GEO LJs and/or LDs detected from optical lightning observations are characterized in detail. The presented figures and Table 4 show that these storms align closely with severe thunderstorms in most mean values and also in the distribution extremes of various thermal, moisture, and dynamical storm characteristics. Specifically, these storms exhibit statistically more organized convection with stronger updrafts, indicated by mean OT counts of 0.2 or higher, BTD of the OT to the cloud shield of 2 K or more, and cold CT temperatures (below 220 K). This contrasts with thunderstorms lacking LJs and LDs, as well as non-severe storms, which typically show no OTs and have CT temperatures above 235 K. In addition, the latter are less likely to produce high amounts of rain and, thus, less likely to cause dangerous flash floods. Findings of this study align well with previous studies, which also reported highly organized convection and higher intensity of convection-related radar variables for thunderstorms with ground-based LJs. (e.g., Chronis et al., 2015; Wapler, 2017; Nisi et al., 2020; Rigo and Farnell, 2022). The GLM multiLJ storms are found as the most organized ones (BTD for OTs average 4.8 K), with the highest CRR (mean of 26.4 mm/h) and potentially the most dangerous thunderstorms. The storms that produce a LD but no LJ have statistically lower CTs (222 K), produce lower CRRs (16.0 mm/h), and weaker OTs (BTD of 0.5 K). However, even these storms significantly surpass the thunderstorms without LDs in all these characteristics (CTs of 237 K, CRRs of 5.3 mm/h, mostly no OTs), meaning the convection is more stable. Some severe thunderstorms, mostly with severe wind reports, did not produce GLM LJs and LDs. These severe thunderstorms are characterized by less organized convection (CT temperatures of 225 K vs. 214 K, OT depth of about 0.5 K vs. 4.6 K), and the maximum CRRs (12.5 mm/h) were also lower than for the severe storms with LJs and/or LDs (24.7 mm/h). Furthermore, it is possible that these storms did produce severe weather that was not reported since severe weather databases have documented limitations (e.g., Hulton and Schultz, 2024; Schroeter et al., 2021)

### 4 Discussion and final remarks

This work had the objective to understand lightning jumps (LJs) and lightning dives (LDs) identified from GLM lightning records. This analysis examines thunderstorm characteristics for storms with and without LJs and LDs, as well as for severe and non-severe thunderstorms. The NWCSAF nowcasting software provides GOES-16 ABI characteristics for tracked thunderstorm cells. Based on the storm flash rate (FR), the FRarea LJ and LD algorithms (Erdmann and Poelman, 2023) were applied to automatically detect LJs and LDs for each thunderstorm trajetory. LJs, LDs, and NCEI severe weather reports allow then the categorization of the thunderstorm trajectories so that TS categories are obtained for LJ and non-LJ, LD and non-LD,

and severe and non-severe thunderstorms. All ABI characteristics can be compared across different categories. To summarize the findings, the questions posed in the introdcution are addressed:

**What do GLM LJs tell us about the storms structure from a satellite point of view?** Thunderstorms with GLM LJs have larger footprint areas compared to those without GLM LJs. Additionally, these LJ storms feature very high cloud tops composed of ice crystals. Thunderstorms with GLM LJs also exhibit above-average overshooting top (OT) counts and depths, whereas OTs are scarcely present in thunderstorms without GLM LJs. OTs result from strong convective updrafts, and in agreement with the OTs, there is evidence in the data that the ice crystals in thunderstorms with GLM LJ are larger than in thunderstorms without GLM LJs. Another import result is the high convective rain rates (CRRs) in the storms with GLM LJs, which are almost 4 times higher than in storms without GLM LJs (summary of values in Table 4). Overall, GLM LJs indicate well organized convective cells that often feature stable convective updrafts.

**Are GLM LJs useful to assess thunderstorm severity?** The thermal, cloud top (CT), moisture, and precipitation characteristics of thunderstorms with GLM LJ were remarkbly similar to the severe thunderstorms (Table 4). In addition, storms without GLM LJs and non-severe thunderstorms agree in the analyzed characteristics. Especially thunderstorms with multiple LJs showed maxima in the OT characteristics and CRRs that were even higher than for the hailstorms and just slightly lower than for the thunderstorms with reported tornadoes. Hence, multiple GLM LJs during a lifecycle of thunderstorm cell are an important indicator of a dangerous storm cell.

It should be mentioned that severe weather is observed in storms without LJs, and that there are non-severe storms that had GLM LJs. Users of the algorithm are advised to be aware of its limitations, especially when using it for operational purposes. The algorithm can indicate the occurrence of severe weather in a thunderstorm, but as with all real-time tools, the user must take into account many other elements, such as signatures observed in radar images, satellite data or terrestrial lightning detection networks. Erdmann and Poelman (2023) analyzed the critical success index (CSI), probability of detection (POD), and false alarm ratio (FAR) for GLM LJ as severe weather predictor. They found a CSI of 0.4 (POD of 0.58, FAR of 0.44) for LJ and severe weather within one storm cell, and a CSI of 0.48 (POD of 0.65, FAR of 0.37) for matching of LJs and severe weather reports that are close in space/time. Studies that used ground-based LJs report CSI of about 0.1 with FAR greater than 0.8 (Murphy, 2017; Miller et al., 2015), POD of 0.69 and FAR of 0.63 (Schultz et al., 2016), CSI of 0.58 (Farnell et al., 2017), POD of 0.45 and FAR of 0.3 (Nisi et al., 2020), and CSI of 0.41 (Tian et al., 2022). The latter two used hail events as reference. Although the concept of GLM LJs is still relatively new, the skill obtained for nowcasting severe weather is similar as in these studies using ground-based lightning observations.

**Do GLM LDs provide additional information about the thunderstorms?** 70 % of the thunderstorms with a tornado also had a GLM LD. In comparison, only 48 % of the tornadic storms also featured an GLM LJ. In total, there were 188 severe thunderstorms with a LD but no LJ. On the contrary, thunderstorms with LDs exhibited deep convection, but LJ storms and severe thunderstorms were statistically more organized (with higher CTs, OT characteristics, and higher CRRs). The number of GLM LDs was twice that of GLM LJs. The applied LD detection algorithm finds LDs for almost 80 % of storms with a FR above the FR activation criterion (i.e., 10 GLM flashes per minute). Hence, the category of

LD storms comprises the majority of storms with sufficiently high FR. A modified LD algorithm could be tested in the future to filter out LDs that occur in dissipating storms.

For the first time, these findings are based on the use of optical LJs detected from GEO orbit. All previous studies used LJs that were identified from ground-based lightning locating systems (LLSs) that detect electromagnetic signals (LF or VHF) rather than optical pulses. It should be mentioned that the results were similar with the use of other LJ and LD algorithms from
Erdmann and Poelman (2023) such as the RIL algorithm. LDs could occur when the storms dissipate and the flash rate (FR) drops naturally due to the dissipation of the storm.

The most important finding of this study remains the behavior of thunderstorms that produced multiple GLM LJs during their lifecycle. These storms feature the strongest updrafts and highest cloud tops, and have all ingredients to produce severe weather and very high rain rates. Especially (though not exclusively) these storms should be closely monitored for weather
advisory and weather warnings. GLM-based LJs have been observed to precede severe weather events by tens of minutes (Erdmann and Poelman, 2023) and may mean the first noticeable signature of developing weather hazards.

*Code availability.* Python 3.8 coding was used, with standard libraries and Matplotlib for the figures. The code was mainly developed during Felix Erdmann's EUMETSAT fellowship and as such is the property of the funders EUMETSAT and RMIB. Python code that is subject to active research and further studies cannot be made available. Parts of the code (Python scripts) are available from the corresponding author
upon request.

*Data availability.* The NWCSAF software is available on the NWCSAF website (https://www.nwcsaf.org). ABI data are available online via NASA EARTHDATA (https://search.earthdata.nasa.gov/portal/idn/search?fi=ABI). GLM data are available online via NASA CLASS (https://www.avl.class.noaa.gov/saa/products/search?sub_id=0&datatype_family=GRGLMPROD&submit.x=22&submit.y=2). Access to ECMWF data requires a user account and access token. The NCEI weather reports are online (https://www.ncdc.noaa.gov/stormevents/).

**Appendix A: GLM flash detection efficiency impact**

GLM performance depends on the nature of lightning itself, and also on cloud characteristics and thunderstorm development. The instrument performance can be assessed through comparison to other lightning locating systems (LLSs) via a relative detection efficiency (DE) that expresses the ratio of lightning processes that are detected by the reference LLS and could also be detected by the evaluated LLS. GLM DE varies with the region within the field-of-view (Cummins, 2021; Blakeslee et al., 2020;
Murphy and Said, 2020; Marchand et al., 2019). Technical aspects like the viewing angle and parallax play a role (Bruning et al., 2019). Furthermore, thunderstorm evolution and cloud characteristics influence GLM performance (Borque et al., 2020; Lang et al., 2020), and GLM DE seems to degrade during periods of overshooting tops (OTs). Zhang and Cummins (2020) reported in agreement with most of the previously cited studies, that GLM performs optimal for large, long lasting flashes. The GLM DE decreases during periods of very high flash rates or small flash sizes. As an optical instrument, GLM shows

430 day-night DE differences: Overall, Cummins (2021); Zhang and Cummins (2020); Murphy and Said (2020); Marchand et al. (2019) suggest 10-15% higher DE at night than during daytime over the CONUS. (Bateman et al., 2021; Erdmann, 2020) found small differences in GLM day- and nightime DE due to the use of coarse criteria and a limited region, respectively. Nevertheless, the influence of GLM flash DE on LJ/LD detection and the results of this study are anticipated to be minimal, as demonstrated in Appendix A1.

## A1 Impact of GLM flash DE on the detection of LJs

The dependency of GLM flash DE on the region is a systematic problem. Therefore, it is possible to analyze GLM observations in regions exhibiting different DE to assess the impact of GLM DE on the outcomes of this study. Based on Cummins (2021), a detection threshold of 3 fJ is used to separate U.S. states with lower (central and northern CONUS) and higher (southeast CONUS) GLM DE. Then, LJs have automatically been detected (Section 2.6) and verified using NCEI severe weather reports.
Figure A1 displays the counts of LJs and NCEI severe weather reports for the region of higher (a,c) and lower GLM DE (b,d), respectively. The pixels of maximum LJ counts agree with the occurrence of severe weather. In some regions, LJ activity is highest where tornadoes occurred (e.g., southern Mississippi or Minnesota). In other regions (e.g., Louisiana) high LJ counts correlate with the local maximum in hail events. The high count of NCEI weather events around the Great Lakes and northeast CONUS mainly comes from wind reports that are less spatially correlated to the LJs compared to hail and tornadoes.
Overall, critical success index (CSI) yield similar skill in both regions when verifying the LJs with NCEI severe weather events (not shown). The correlation of LJs to NCEI reports does not depend on the different GLM flash DE. However, it was found that the number of false alarms, i.e., LJs that occurred independently of a severe weather events, could be reduced in the region of higher GLM DE if the LJ detection algorithm uses a higher FR threshold than for the full CONUS (see Section 2.6). It should be mentioned that this study considers the occurrences of LJs, not their strengths. LJ strengths and maximum flash
rates may well be higher in the region of higher GLM flash DE, however, the number of LJs and their correlation to NCEI reports was little affected by the GLM flash DE.

## A2 Thunderstorm cloud characteristics

There were, in total, 16155 and 8818 thunderstorms in the region of higher and lower GLM DE, respectively. Both regions contain a statistically relevant number of cases to analyze and compare the thunderstorm cloud characteristics. In particular,
this section examines the characteristics of thunderstorms and investigates whether storms with LJ and/or LD exhibit distinct characteristics in the two regions. Main differences in the cloud characteristics occur due to the climatology (e.g., average temperatures in regions, the tropopause height) and for geographical reasons (e.g., moisture from the Gulf of Mexico). For example, Figure A2 presents the BTs of the ABI IR12.3 channel for (a) the region of higher GLM DE and (b) the region of lower GLM DE. Brightness temperatures (BTs) are on average about 2 K colder in Figure A2(a) than in Figure A2(b), meaning
the CTs reach higher altitudes. Figure A3 compares the WV6.2 channel for the region of (a) higher and (b) lower GLM DE. Again, the BTs in the region of higher GLM DE are about 2 K colder than in the region of lower GLM DE. The water vapor channel gets saturated at higher altitudes in the region of higher GLM DE as the atmosphere contains in general more moisture

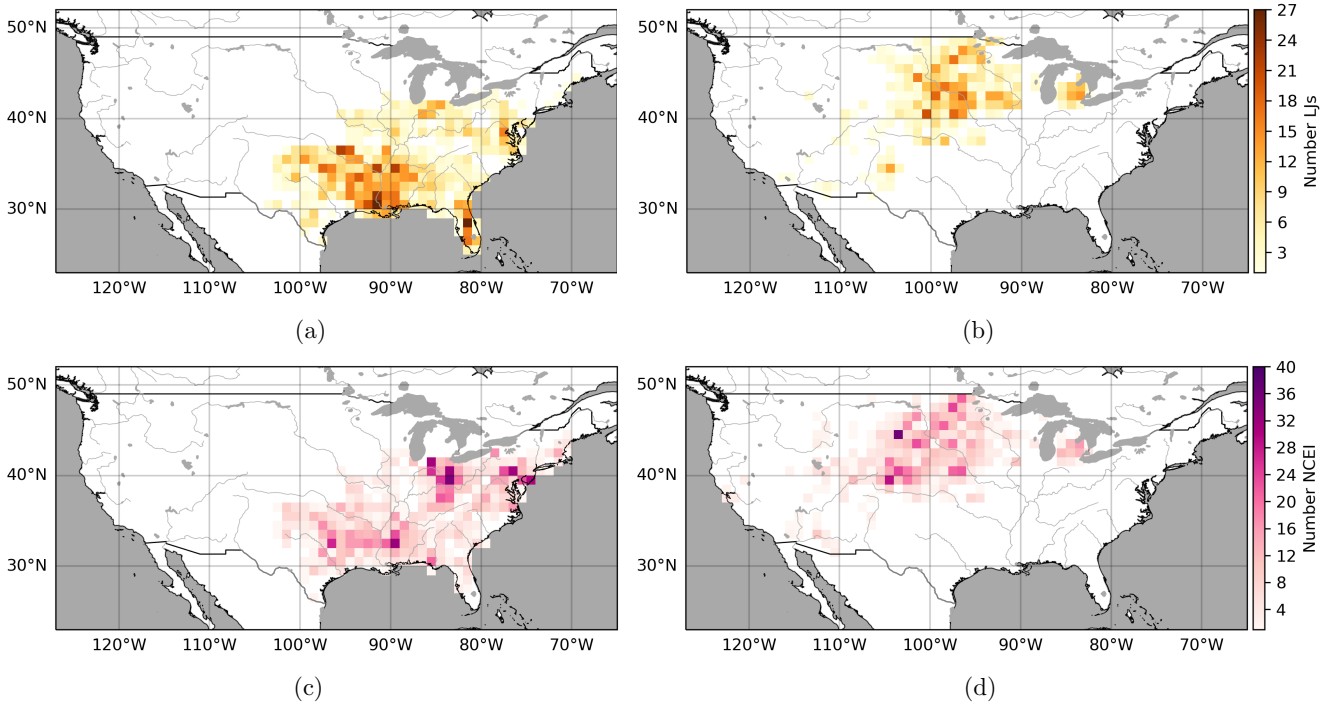

**Figure A1.** Number of (a,b) LJs, and (c,d) NCEI weather events (tornadoes, hail, wind) per $1° \times 1°$ pixel in the region of (a,c) higher and (b,d) lower GLM DE.

than in the region of lower GLM DE. The WV7.3 channel results confirm the presented finding also for the mid-level water vapor. These differences can be observed throughout all the TS categories (Table 3) and, thus, they are independent of the LJ/LD 465 detection. A detailed analysis of the TS categories withLJ and withLD in the two regions confirmed that the thunderstorms with LJs and those with LDs, respectively, feature similar characterisitics when the climatology bias is corrected. The thunderstorms in the region of higher GLM DE are on average smaller than in the region of lower GLM DE, indicating that the storm types differ and there are likely more single-cell, thermally driven thunderstorms in the southeast than further north in the CONUS. It is also known that large, long-lived thunderstorms or mesoscale convective systems can form along air mass boundaries in 470 the Great Plains, and this region is mainly contained in the region of lower GLM DE. OT counts (0.31 and 0.25 in the region of higher and lower GLM DE, respectively) and OT DT max (2.8 K and 2.6 K, respectively) show little variation in the two regions for severe thunderstorms. On the other hand, OTs occurred almost twice as often in the region of higher GLM DE than in the region of lower GLM DE for the LJ (mean counts of 0.48 and 0.28) and LD storms (0.28 and 0.16). OTs were also deeper in both LJ (DT max of 4.3 K) and LD storms (DT max 2.5 K) in the region of higher than in the region of lower GLM 475 DE (3.0 K and 1.7 K for LJ and LD storms, respectively). The region of higher GLM DE is prone to see thunderstorms that develop rapidly, have high flash rates and strong updrafts. These features are typical for supercells that are more frequent in the region of higher then in the region of lower GLM DE (e.g., Ashley et al., 2023; Thompson, 2023).

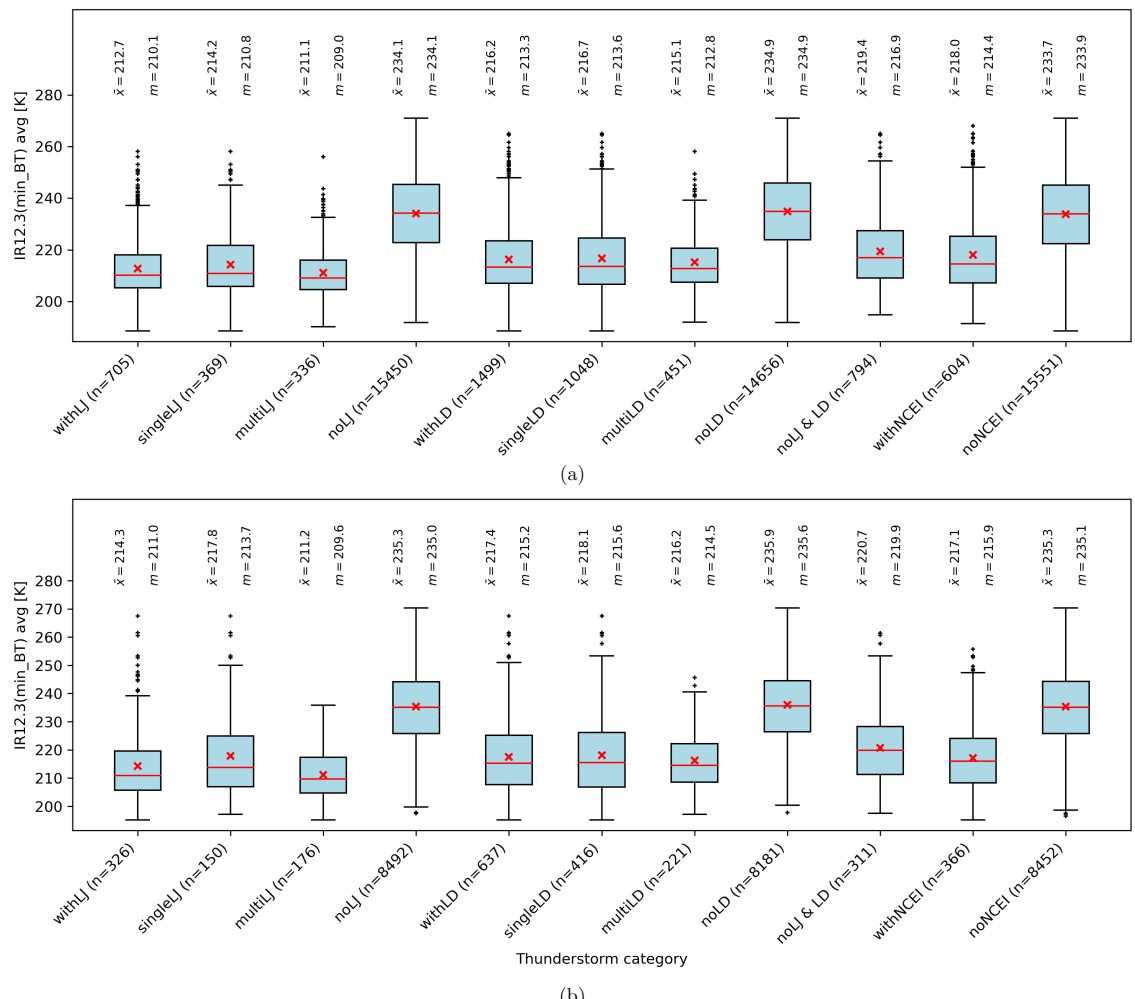

**Figure A2.** Trajectory minimum over cell-averaged BTs of the IR12.3 ABI channel for the region with (a) higher and (b) lower GLM DE. $\bar{x}$ shows the mean, $m$ the median for each TS category.

## A3 Appendix Conclusion

LJ and LD detection detection algorithms could apply higher FR thresholds to reduce the number of false alarms in the region of higher GLM DE. Nevertheless, even then the overall CSI skill for LJs/LDs as severe weather predictors remains similar as in the region of lower GLM DE, as fewer hits are generated when applying higher FR thresholds. Hence, LJs and LDs can be detected using the same algorithm type over the entire central and eastern CONUS without a significant impact on the algorithm performance. In addition, a given thunderstorm characterisitic changes from the higher to the lower GLM DE region for all TS categories to the same extent as for the thunderstorms with LJs and/or LDs. Hence, observed differences in

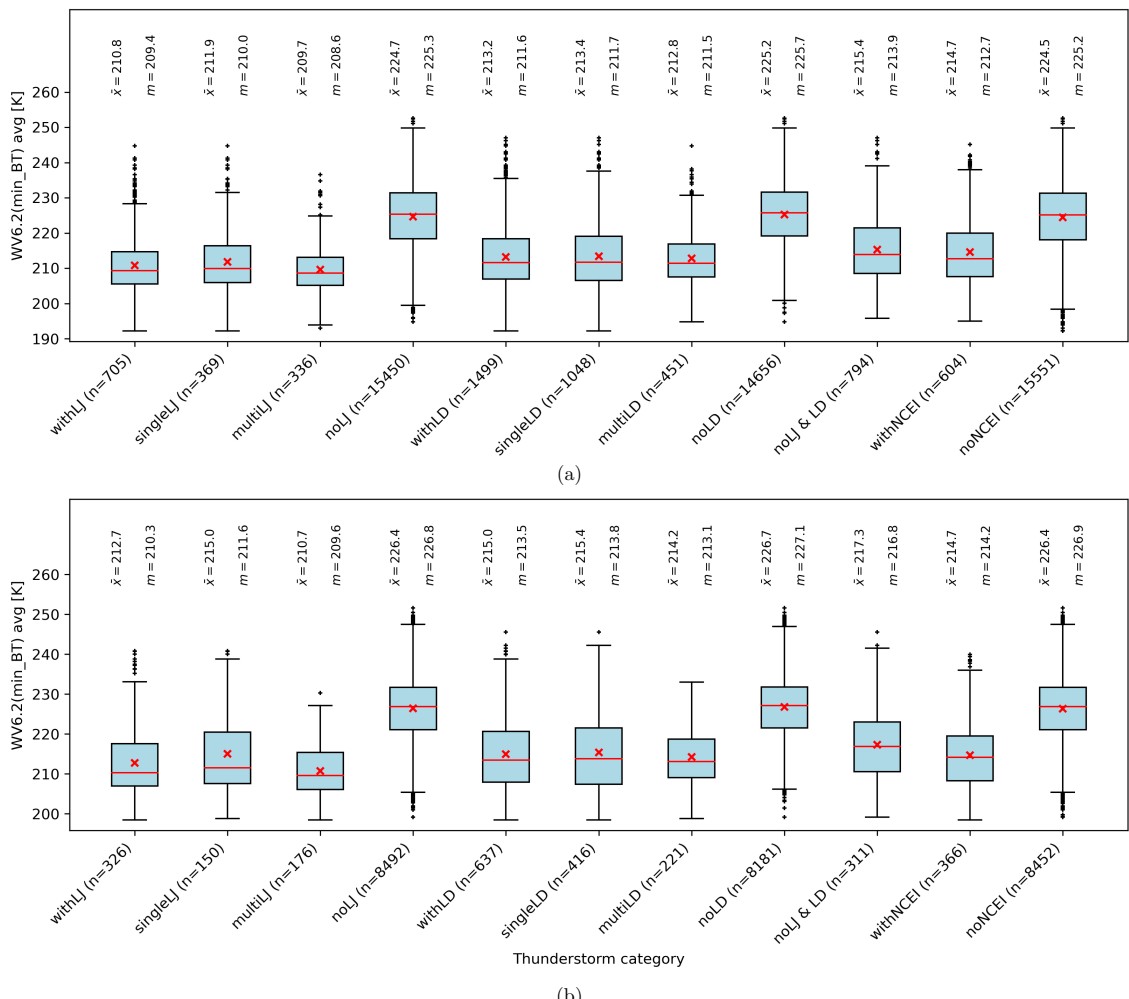

**Figure A3.** As Figure A2 but for BTs of the WV6.2 ABI channel.

the thunderstorm characteristics are mainly attributed to the different climate and weather conditions in the southeastern and the remaining CONUS.

*Author contributions.* FE wrote the paper text and created the figures. DRP was involved in content creation and internally reviewed the paper prior to submission. Both contributed to the journal peer review process.

*Competing interests.* The authors declare that they have no conflict of interest.

*Acknowledgements.* The work of FE was supported by the fellowship "Towards an automated severe weather warning tool based on MTG-LI and FCI data" from the European Organisation for the Exploitation of Meteorological Satellites (EUMETSAT). The hosting institution of this fellowship is the Royal Meteorological Institute of Belgium (RMIB). The authors thank F. Autones, M. Claudon, and J.-M. Moisselin for providing, before an official release, the NWCSAF software package with included GLM data reader and their expertise on the RDT-CW package. The authors thank N. Clerbaux for setting up the new software version of NWCSAF at the RMIB and for downloading

necessary satellite data. The authors acknowledge ECMWF for providing the necessary NWP data. Finally, 3 anonymous referees reviewed this manuscript to improve it an to obtain it in this published version.

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
