# Peer review of "Insights into Thunderstorm Characteristics from Geostationary Lightning Jump and Dive Observations"

_EGUsphere, 2024_

## Author Comment (AC1)

We thank the anonymous referee #2 for his/her review and the positive feedback for our manuscript. We will address the referee comments in blue font below.

**RC2**: 'Comment on egusphere-2024-174', Anonymous Referee #2, 25 Mar 2024

Title: Thunderstorm characteristics with lightning jumps and dives in satellite-based nowcasting

Author(s): Felix Erdmann and Dieter Roel Poelman

MS No.: egusphere-2024-174

MS type: Research article

Thunderstorm characteristics with lightning jumps and dives in satellite-based nowcasting" by Felix Erdmann and Dieter R. Poelman presents an attempt of adapting the Lightning Jump to satellite imagery (Lightning Imager), as also introduces a new concept, such as the Lightning Dive.

We agree that the Lightning Dive (LD) is a new concept that has not been found in any publication (hence, no reference). The idea was first mentioned by US colleagues during the GLM science meeting 2022 (remotely), that found decreases in the lightning activity of a storm prior to or during tornadoes. The LDs are defined as negative LJs.

Since the objective of this paper is to investigate satellite-detected changes in the flash rate (i.e., GLM LJs and LDs), we included the LDs in our study. We added a paragraph in the introduction that introduces the idea behind the LDs:

"The LD exhibits behavior contrary to that of a LJ, leading to a rapid reduction in the FR as first mentioned by Losego et al. (2022). It is based on the idea that a decrease in lightning activity can precede events such as tornadoes or significant hail. That is the case since the rear flank downdraft (RFD) can be related to tornado development (e.g., Satrio et al., 2021; Mashiko, 2016; Markowski, 2002). Within the RFD, internal momentum surges can temporarily weaken the updraft or alter the hydrometeor content. Such a weakening of the updraft is correlated with reduced lightning activity, as noted by Deierling and Petersen (2008). Furthermore, downdrafts caused by intense rainfall or hail can interact with the storm's updraft and charging structure. These interactions can temporarily reduce lightning activity, as fewer ice particles collide, which is necessary to sustain strong electric fields through non-inductive charging."

The results are interesting and promising, having in mind the put in operation of the MTG in an early future. The main issue the lack of information regarding this last concept, because it is a novelty and it should be better introduced, with some examples that should help to make more understandeable to the reader.

Thank you for the positive feedback. We introduce the idea behind the LDs in the revised paper, but simultaneously focus more on the LJs.

Besides, there is some vaguety about the "severe weather" convept, which it must be better explained.

Severe weather is defined by the presence of a tornado, strong winds, and/or large hail. We use the NWS definition of a severe thunderstorm ("A thunderstorm that produces a tornado, winds of at least 58 mph (50 knots or ~93 km/h), and/or hail at least 1" in diameter.", https://www.weather.gov/bgm/severedefinitions). We added this reference to the introduction in manuscript.

Finally, the results are sometimes presented in a difuse way and I think the Authors could do an effort to improve the quality of presentation.

We re-structured our manuscript, with more focus on specific selected cloud characteristics.

You can find below the specific comments associated with the review.

L21: "The opposite behavior, a sudden decrease in the FR is termed a lightning dive (LD)" Referència?

We introduce the term here, based on an idea of colleagues from Georgia Tech. We introduce the concept in a short paragraph in revised manuscript, and we added the link to the presentation (https://goes-r.nsstc.nasa.gov/home/sites/default/files/2022-10/session_4/Losego_Megagraph.pptx, last accessed 19/04/2024) and the meeting (https://goes-r.nsstc.nasa.gov/home/index.php/meeting-agenda-2022)

Introduction: "Severe" or "Adverse" weather?

We added that we use the NWS definition of a severe thunderstorm.

Have you noticed if "However, most LJ algorithms were tuned based on ground-based lightning mapping array (LMA) data." are operational and running in real-time? (L44)

For example, the Meteorological Service of Catalonia runs a LJ algorithm operationally (Rigo and Farnell, 2022). A real-time use is possible since the algorithm is fast (less than 1 minute delay). Since the LJ algorithm are defined on the flash level, it should be considered that ground-based data must be clustered to flashes before the application of the algorithms, and there are effective algorithms to do that.

L80: "The object-oriented approach can effectively differentiate between convective and non-convective cloud cells, and track the convective cells through image recognition, identification of known patterns, and statistical models." Have you any feed-back comparing with weather radar database?

No, weather radar data were not analyzed and we do not know any comparison of the RDT cell tracking algorithm with a radar based cell tracking approach. Usually, the RDT cloud cells would

be somewhat larger than radar based cells since the cloud shield covers a larger area than the convective cores and/or updrafts that can be detected with weather radars. You can compare Table 2 of Schultz et al. (2016) and Table 2 of Erdmann and Poelman (2023) for more details.

L104: why these thresholds?

We used the thresholds that were defined in previous studies (Schultz et al., 2009, 2011), also to be able to compare our results to the results of those studies.

Caption: "Figure 1. Relations between tools and data of this study." Is this a data flow? If yes, please change the caption.

No, the figure does not show a data flow. It simply introduces the different data types and software packages. We adapted the caption: "Data and product types of this study. The dependencies of products can be read from the top to the bottom, and they are also indicated by the arrows. On top, there is the input (grey), boxes with colored frames indicate the intermediate products, and the features in colored boxes are analyzed in the Results section."

L 90: "During our selected study days (Table 1), there was one important GOES-16 downtime from 03 Jun 17:00UTC to 04 June 01:30UTC." change to "It is worth noting that there was one relevant GOES-16 downtime from 03 Jun 17:00UTC to 04 June 01:30UTC (Table 1)."

OK

L92: "It should be noted that only thunderstorms are analyzed that are defined as RDT cloud cells with GLM lightning activity." This sentence is difficult to understand.

We re-wrote the sentence: "An RDT cloud cell with matched GLM flashes defines a thunderstorm."

L93: "This studies aims at understanding the meaning of LJs and LDs for thunderstorm characteristics." -> "This study aims to understand the meaning of LJs and LDs for thunderstorm characteristics."

OK

L95: "Such cells generally give rise to weaker weather phenomena compared to major thunderstorms." Partially disagree: warm rain clouds cause heavy rainfall and flash floods in many regions around the World.

Indeed, heavy rainfall or flooding can have other causes than a thunderstorm. We removed the sentence from the manuscript as it is not essential for this study.

Section 2.3: Have you considered the parallax effect? Besides, which is the general size of a RDT cell? Have you manually evaluated these thresholds in any case, to validate them?

Yes, the NWCSAF software corrects for parallax. The average size of an RDT cell is about 619.9 km2, and 2988.5 km2 for thunderstorm cells (Erdmann and Poelman, 2023). Please note that most of the time (>90% of NCEI reports matched to an RDT cloud cell) the report is located within the cloud cell contours. The 50 km threshold is applied to account for an uncertainty in the report location and/or the defined cloud contour. If a report is located within 50km of the cloud cell contour, then (i) the report is considered only if it isn't within the contour of another cloud cell, (ii) only the closest cloud cell is matched to the report. Hence, the 50 km are in practice very variable and state the maximum distance to match cell and report. We did not test longer distances.

We added to line 84: "RDT also corrects for satellite parallax effects."

L120: Have you evaluated the limitation of using standard scan instead of rapid scan? Which is the time running of the NWCSAF software?

No, we didn't. The processing times and data volume would increase significantly. This could be tested for a limited region. Since we tried to include many storms in the CONUS and GLM-16 field of view, we decided to use the 10-minute update cycle. NWCSAF can run with any satellite data that is given to the software as input. Our NWCSAF runs include the analysis and a 60-hour forecast. For one 24h period, NWCSAF runs about 20 hours on our institute server. However, a single 10-min slot with many active cells can need a processing time of up to 3 hours on our server. It should be possible to reduce the processing time through deactivation of some NWCSAF modules, but we did not test this.

L160: I think you should present an example of the LD, because is not an usual phenomenon as LJ and it needs to be assimilated by the reader.

We added a paragraph to explain the meaning of LDs in the introduction. LDs are really the opposite of the LJs where the flash rates decrease significantly (as of the LD detection algorithm) and rapidly from one minute to the other.

Section 3.1: the thunderstorm' categorization should be presented as part of the methodology. The analysis of the categories should be included in the Results section.

OK, we considered these suggestions for re-structuring the revised manuscript.

L172: Why do you think there are more LD than LJ?

This is based on the current algorithm configuration. LDs are often not only detected during the development of the storms, but also during their decay. A modified LD algorithm could exclude the dissipation phase of the storm when lightning activity always decreases and lightning dives can be triggered randomly. For example, require the flash rate to increases again after a sudden drop was detected.

L173: "A thunderstorm can produce more than one type of severe weather (the sum of withTornado, withHail, and withWind is greater than the number of severe TSs)." Please, write more clearly.

New: "The 3 categories withTornado, withHail, and withWind include the thunderstorms that produced tornadoes, hail, or wind, respectively. The sum of their counts (79+438+645=1162) exceeds the number of severe thunderstorms (970) meaning several thunderstorms produced more than one type of severe weather."

L174: This sentence is reduntant and can be shortened

OK. We removed the explanation and just kept the first part: "All storms with a LJ also had a LD."

L176: "There are storms with LJs and/or LDs that did not produce severe weather (59.9 % and 71.9 %, respectively)." Disagree. Severe storm can occurr but it has not been reported (or reported to the used database)

That's a good point. We changed the sentence: "There are storms with LJs and/or LDs where no severe weather was reported (59.9 % and 71.9 %, respectively). However, these storms might have produced severe weather that was not reported."

L319 Rigo and Farnell (2022) did not analyze LMA-based multi-LJ storms

Thank you for the comment. They used the XDDE LLS, not an LMA. We replaced "LMA-based" by "ground-based" in the manuscript.

---

## Author Comment (AC2)

We thank the anonymous referee #1 for his/her review of our manuscript. We will address the referee comments in blue font below.

RC1: 'Comment on egusphere-2024-174', Anonymous Referee #1, 13 Mar 2024

I have read this paper on lightning jumps and dives for the use in nowcasting, and while the topic is interesting, I think the paper needs major revisions before it can be considered for publications.

**Major comments:**

1) The paper is much too long. It appears to be a follow on of a previous paper by the authors that shows similar results.

We agree that the original manuscript is very long. We have revised the manuscript in a way we believe has improved the readability. Yes, the manuscript follows on the work of our previous publication where the lightning jump algorithm has been tuned for the GLM instrument. However, the objective and also results of this new manuscript have nothing in common with the previous paper. Here, we analyze cloud characteristics, while the previous paper correlated LJs and severe weather reports in order to optimize the LJ algorithm parameter for the GLM.

The paper appears to be a shopping list that brings basically all parameters possible to compare with lightning jumps and dives, without the focus needed and defined by the title of the paper. This is very distracting for the reader since there is too much information provided without a clear storyline. If you want to use lightning for nowcasting of severe weather or floods, you should focus on that.

We propose to slightly change the title into "Thunderstorm characteristics with satellite-based lightning jumps and dives" which better reflects the content of the manuscript. The main objective is not the nowcasting of severe weather (as in our previous paper) but to understand the LJs and LDs and what they reveal about the thunderstorms that produces them. To this end, we include a variety of thunderstorm characteristics and compare them for thunderstorms with and without LJs (and LDs).

It is clear to all (nothing new) that storms with LJs with be more developed with stronger updrafts, higher tops, colder tops, more overshooting turrets, etc. This is not new, and hence does not contribute to our scientific knowledge. Just presenting these results again does not make them novel or innovative.

This paper uses LJs and LDs that are detected from GLM observations. GLM observes lightning in the optical (oxygen band). Hence, the detection of lightning captures different processes of the lightning discharge that are different from those observed by VHF LMAs. Although the term LJ is used, the GLM LJs are often very different from the LJ that one would detect in LMA data, for example Murphy and Said (2020) showed that the GLM LJs are less correlated to radar

variables than LMA LJs. The optical GLM LJs are apparently less correlated to cloud microphysics than the LMA LJs since GLM lightning detection is always influenced by additional aspects like viewing angle, cloud optical thickness, scattering of the light. That is exactly the point of this study. We want to find out if the optical GLM LJs and LDs correlate to cloud characteristics that are typical for severe storms although there are these additional aspects that affect the detection from space.

So I would focus ONLY on the use of LJ for severe events. Remove all the analysis not related to severe weather and LJs

That has been done in our previous publication and is not the objective of this work. Here, we investigate whether the LJs and LD are found for storms with similar characteristics as severe thunderstorms. This is done for the satellite-based GLM LJs and GLM LDs. The latter have never been studied before and it is one main objective to find out whether this concept provides meaningful information.

2) This brings me to the next point that as the authors point out in lines 176-177, 60% of storms with LJs do not produce severe weather, and there are severe weather events without LJs in 57% of cases. So this shows that lightning jumps from the GLM are NOT good for predicting and nowcasting of severe weather. So why continue with the paper then.

We do not agree with the referee here. If LJs help detecting 1 in 2 severe thunderstorms in advance, then this is a meaningful information for nowcasting severe weather. As referee #2 pointed out, the percentage is also affected by the fact that not all severe weather could be reported and the 59.9% is likely too high.

Note that there is no perfect nowcasting tool. Forecasters always combine different datasets (e.g., radar, satellite, and lightning) to identify the dangerous thunderstorms. In addition, the GLM LJs are based on total (CG+IC) lightning detection, and Erdmann and Poelman (2023) found that the leadtimes before the severe weather event can be longer than with other nowcasting tools (i.e., several tens of minutes). This is because the GLM detects the early high IC flashes well, and a GLM LJ can arise from that early lightning activity during the thunderstorm development.

Either the paper is not about nowcasting, and then you need to change the title and focus of the paper, or you need to prove that LJs are good in predicting severe weather.

Ok, we removed nowcasting from the title of the paper. The new title is "Thunderstorm characteristics with satellite-based lightning jumps and dives". Please note that lines 49-50, and lines 57-58 clearly state the objective of this paper mentioning optical LJs and LDs.

Here I would separate the analysis for tornadoes, wind damage, and hail. And if LJs are not good for detecting severe weather from GLM, then that too is a result, even if "negative". But no need to go on and on about cloud parameters linked to thunderstorms with LJs.

This would be a different study. We performed parts of the suggested tasks during the work on our previous paper (Erdmann and Poelman, 2023), and we did not see a specific behavior or correlation of the GLM LJs to a specific severe weather type.

3) I do not understand the interest in lightning dives (LDs). This is the first time I hear of their "importance" as a measure of thunderstorm activity. The physical meaning of LJs is the intensification of the storms, with stronger electrification, more rainfall, and maybe more severe weather. But why should we be interested in LDs which imply the decay of the updrafts in the storm, the drop in electrification, the drop in lightning, and hence the drop in probability of severe weather. Why should LDs be important for severe weather. Please explain the physical connection if you plan to keep talking about it. I would focus only on the LJs and remove the LDs analysis.

We added a paragraph to introduce the concept of the LDs and potential meaning. Based on previous preliminary results of US colleagues, the rear flank downdraft (RFD) can temporarily weaken the updraft and reduce the lightning activity shortly before or during a tornado. The RFD can also cause severe winds at the surface (reported events). Since LDs have never been studied, we included them in this work to investigate their meaning. We agree that overall the LJs are the more interesting features. We modified the conclusions to state this more clearly.

The following paragraph was added after line 27: "The LD exhibits behavior contrary to that of a LJ, leading to a rapid reduction in the FR as first mentioned by Losego et al. (2022). It is based on the idea that a decrease in lightning activity can precede events such as tornadoes or significant hail. That is the case since the rear flank downdraft (RFD) can be related to tornado development (e.g., Satrio et al., 2021; Mashiko, 2016; Markowski, 2002). Within the RFD, internal momentum surges can temporarily weaken the updraft or alter the hydrometeor content. Such a weakening of the updraft is correlated with reduced lightning activity, as noted by Deierling and Petersen (2008). Furthermore, downdrafts caused by intense rainfall or hail can interact with the storm's updraft and charging structure. These interactions can temporarily reduce lightning activity, as fewer ice particles collide, which is necessary to sustain strong electric fields through non-inductive charging."

**Minor comments:**

Title: The paper does not focus on nowcasting. Either it should, or the title should be changed.

We removed "nowcasting" from the title as it is not the objective of this paper. New title: "Thunderstorm characteristics with satellite-based lightning jumps and dives"

line 19: ...certain maxima and minima

OK

line 23: do you have a reference for "lightning dives" other than your own papers? Please add reference

No, we do not have a reference. We added a short paragraph to introduce the concept why it could be interesting to study it.

line 93:  study aims

OK

line 124: Problem with text after 777.4nm

Thank you. We corrected the sentence. It was a formatting issue.

line 152: Both algorithm types use a FR

OK

line 172: do not produce

OK

line 174:  It is obvious that what goes up must come down.  Hence all LJs with be followed by a LD.  Is this not a trivial conclusion

We understand the concern raised regarding the perceived triviality of the conclusion. However, we believe the conclusion is more nuanced than it may seem. The decrease of the flash rate can also happen slowly, and then one would not identify a LD there. We think that the LDs often occur during the dissipation phase of thunderstorms but further research is needed to confirm that idea. A modified LD detection algorithm could look at the development and mature phases of the storm only. For example, require the flash rate to increases again after a sudden drop was detected.

line 179: would still show

OK

line 196:  Again, it is obvious that min pressure implies higher cloud tops and min BTs

We thought it is worth mentioning that the characteristics are consistent. However, the revised manuscript tries to reduce redundant information in order to be more concise.

Figure 2 appears to be a shopping list with no clear point.  Are the overlapping blue boxes in 2a significantly different from each other?

We agree that Figures 2 to 4 are hard to read. The figures should introduce all thunderstorm categories and all cell characteristics that were studied. Since we have this information in Tables 2 and 3, we decided to simplify the figures. The revised manuscript discusses and shows

selected thunderstorm categories and cell characteristics, and excludes/combines the ones that lead to similar conclusions.

If large parts of the IQRs (blue boxes) overlap in Figure 2, then these distributions for the categories are similar for the shown characteristic. If both the quantiles and mean values are higher in one category than in the other one, this can indicate that higher values are more likely in the first category than in the latter although the distributions are rather similar. The 6 (3 in the revised manuscript) categories with the lowest IQRs are statistically different (i.e., only upper outliers of their distributions match the lower distribution outliers of the other categories) from the remaining thunderstorm categories.

Line 217 and 236:  Reference to Fig 2a should occur before Fig. 2b

OK. We modified Figure 2 to have the CT phase in a and the CRRs in b.

Line 330:  Are these conclusions new?  It appears a logical conclusion of more lightning in thunderstorms that has been studied for decades.

Yes, this is new as we are looking at optical LJs detected from GEO orbit. All previous studies used LJs that were identified from ground-based lightning locating systems (LLSs) that detect electromagnetic signals (LF or VHF) rather than optical pulses.

We state this explicitly in the Conclusions, and we added "GLM" to the bullet point heading to clarify that these conclusions mean the LJs/LDs detected in GEO GLM lightning time series.

line 334:  "severe storms often feature".....quantify this. How often?  This qualitative conclusion is not scientific

We did not analyze storm by storm, so we cannot quantify the statement. Hence, we modified the conclusion: "The means, medians, and IQRs of cell characteristic distributions for severe storms resemble those for the storms with LJs (and LDs)."

line 340: "might cause flash floods" is speculation.  Do you not have information of floods at the surface?  If not you cannot speculate so broadly.

We believe it is common understanding that heavy rainfall, in particular high rain rates (defined as large amounts of rain in a short time frame) are the key weather phenomenon causing flash floods. Of course, there are other aspects like the surface type, terrain, runoff that play a role, but it would be beyond the scope of this study to include a full hydrological analysis for each thunderstorm.

line 341:  If the results here are similar to your previous publication, why do we need another publication saying the same thing?  You need to provide new knowledge to advance the sciences and field of thunderstorm research and nowcasting.

Our previous paper did not look at cloud cell characteristics so this manuscript presents new results.

line 347: 2.1 +- what standard deviation? Same for 1.9. Are these values statistically different?

The ratios are calculated from fixed numbers, i.e., the number of LJs and LDs. There is no uncertainty assessment.

We cannot answer whether there is a significant difference between the warm and cold season. We simplified the sentence as follows, as it did not add value to the paper: "During all seasons, the number of LDs is about twice that of LJs."

line 353: tropopause

Thank you, we corrected it.

Line 396: I would like to see more spatial plots in the paper like A1. Maybe a plot showing lightning jumps compared with severe weather reports of different kinds. I would add spatial plots to the main text

We agree that these plots are interesting for nowcasting aspects. However, as the focus of this manuscript is on the statistical analysis of the cloud cell characteristics, we would not add spatial plots to the main text.

Please see the spatial plots for the individual severe weather types in Fig. R1. We can see the locations of LJs best correlate with locations of tornadoes and hail reports, and not always with the locations of wind reports.

[Figure]

Figure R1: Number of (a) LJs, (b) tornadoes, (c), hail, and (d) wind events as reported in the NCEI archive per 1° x 1° pixel.

---

## Referee Report (RR1)

Title: Thunderstorm characteristics with lightning jumps and dives in satellite-based nowcasting

Author(s): Felix Erdmann and Dieter Roel Poelman

MS No.: egusphere-2024-174

MS type: Research article

I appreciated your answers to my comments and the manuscript modifications. The paper has been significantly improved now and I do not have any new major comments.

---

## Referee Report (RR2)

**Review of Erdmann and Poelman paper on Lightning jumps and dives**

This is my second review of this paper, and I find not much has changed in the paper. From the title it appears that the focus on the paper is on how lightning jumps and dives may be used to supply nowcasts of severe weather (hail, tornadoes, wind, etc.). However, the authors themselves show that ~60% of all lightning jump storms are NOT associated with severe weather (~70% are not for lightning dives). Hence, their own analysis shows that using the GEO lightning data we CANNOT use such parameters for nowcasting of severe weather. This should have been the conclusion of the paper – that lightning jumps/dives based on satellite lightning data do not show skill in forecasting severe weather. This negative answer is also important to know.

But then the authors go on a fishing expedition to look for connections between lightning jumps/dives and many other cloud parameters, weather parameters, with no real goal of how to use these relationships. I remind the authors that the journal is about natural hazards and hence the extensive analysis NOT related to severe weather may be better suited for another journal. I would suggest reducing the number of parameters analysed, and to focus on a key research question. Can lightning from GEO be used to supply early warnings of severe weather. Yes or no.

The authors use qualitative descriptions of their results, while for scientific evidence we need quantitative numbers. For example, "overall similar characteristics" (line 8), "can be useful for nowcasting" (line 190), "found to resemble" (line 320), "potentially the most dangerous" (line 324), "resemble" (line 342, 346)

Table 3 shows that most severe weather occurs WITHOUT LJ, while most LJ would give a false alarm if used for warnings of severe weather. Hence, there is no benefit (and maybe even a danger) in using LJ/LD in nowcasting for severe weather warnings. Why not state this?

Figure 1: How can we compare parameters when the normalization hides the actual values? Line 203 states the thunderstorms with LJs cover larger areas than storms with LDs. How can we see this from normalized areas? The min and max values will be different for each subset of data.

Most of the analysis of the cloud parameters is not new, although the lightning data set is new. We know all the connections between severe weather and cloud top, ice, overshooting tops, updrafts, etc. Why another paper on this?

In conclusion, I think this paper is very weak in results, and needs either major revision with a new focus, or should be referred to another more appropriate journal.

---

## Referee Report (RR3)

Dear Authors,

Thank you very much for your detailed answers and the notable improvement of your manuscript. The new version shows a better organization of your Research and provide more curated information about your findings.

Please find below my new comments, some of them appeared from your previous answers (which are marked in bold and between quotation marks):

**"We would like to point out that the lightning jump algorithm in use was adapted to the GEO lightning sensor and is described in detail in Erdmann and Poelman (2023). It does not use the same configuration as the LJ algorithms for terrestrial lightning location systems (LLSs). The commonly used algorithms was developed for lightning mapping array (LMA) with radar-based storm tracking. Erdmann and Poelman (2023) tuned the algorithm parameters to be used with GLM lightning observations and satellite-based storm cell tracking."**

I fully understand your comment, but after carefully reading the two manuscripts (the presented one and the 2023 one), I think that you should include a graphic which could help the reader understand the differences between the original and the proposed version of the algorithm. This scheme should present the life cycle of a thunderstorm from the point of view of the lightning activity and how you transform in the sigma variable and when a jump or a dive occurs.

**"Why is the size of the storms important?"**

The difference in size between supercells or normal cells is quite evident and is an indicator of the intensity and wide of the updraft. This is only one example, but there are many more. For example, large squall lines or MCS tend to last more and to produce more severe weather reports, because the organization is higher and helps to extend the duration. These are two examples of the importance of the size. For sure, there are cases where the size does not provide any relevant information, but these examples are less associated with convection.

In fact, you say in your manuscript that: "the mean cell areas in Table 4 confirm the previous finding. On average, severe thunderstorm cells covered an area of 12,812 km2 (median 4,089 km2). Storms with LJs had an average area of 15,780 km2 (median 6,995 km2), whereas while non-severe thunderstorms and those without LJs typically covered about 2,000 km2 on average, with medians around 550 km2)."

About the list of acronyms, I know that is not mandatory, but in your case can result very helpful to the reader, because the large number of items. It is your choice.

Some new comments:

- Typo in caption of table 4: "Torndao" should be "Tornado"
- I suggest merging subsections 3.1.2 and 3.1.3, reducing part of the text. There a lot of results and I suggest you that focus on the main items.
- You say that "It should be mentioned that severe weather is observed in storms without LJs, and that there are non-severe storms that had GLM LJs". Fortunately, there is not an algorithm able to reproduce the exact behavior of a thunderstorm. Furthermore, as you also indicate in the manuscript (and we previously discussed), there are limitations on the direct ground observations of severe weather. Because of this, I suggest that you replace "Hence, the GLM LJs should not be used as standalone severe weather warning tool but in combination with other data." By "We recommend the algorithm users to consider its limitations, in special at the time of applying for operational purposes. The algorithm can indicate the occurrence of severe weather in a thunderstorm but, as occurs with all the real-time tools, the user must consider many other elements, such as signatures observed in radar imagery, satellite data or terrestrial lightning detection networks."

---

## Author Response (AR2)

Review of Erdmann and Poelman paper on Lightning jumps and dives

Please note the authors' comments on the review highlighted in green.

This is my second review of this paper, and I find not much has changed in the paper. From the title it appears that the focus on the paper is on how lightning jumps and dives may be used to supply nowcasts of severe weather (hail, tornadoes, wind, etc.).
We regret that the significant changes made to the revised manuscript may not have been fully recognized or adequately valued.

We would like to clarify that the current title "Insights into Thunderstorm Characteristics from Geostationary Lightning Jump and Dive Observations" accurately reflects the focus of our study, which is to characterize storms that have GLM Lightning Jumps (LJs) and Lightning Dives (LDs). We would like to stress once more that the manuscript does not address the application of LJs and LDs in nowcasting.

However, the authors themselves show that ~60% of all lightning jump storms are NOT associated with severe weather (~70% are no for lightning dives). Hence, their own analysis shows that using the GEO lightning data we CANNOT use such parameters for nowcasting of severe weather.
As already mentioned, this study does not address the usability of LJs/LDs for nowcasting. That topic was thoroughly discussed in Erdmann and Poelman (2023).
In any case, the reviewers' comment would be valid if the goal were to use GLM LJ as a standalone severe weather warning tool. However, this is hardly ever the case. The information provided by GLM LJs is meant to complement existing tools, such as radar data.
This should have been the conclusion of the paper – that lightning jumps/dives based on satellite lightning data do not show skill in forecasting severe weather. This negative answer is also important to know.
The authors showed in the previous paper (Erdmann and Poelman, 2023), that the GLM LJs, as a predictor of severe weather, reach a CSI of 0.4 (POD of 0.58, FAR of 0.44) for LJ and severe weather within one storm cell, CSI of 0.48 (POD of 0.65, FAR of 0.37) for matching of LJs and severe weather reports that are close in space/time.
We can compare these results with studies using ground-based LJs for detecting severe weather:
- Schultz et al. (2016, doi: 10.15191/nwajom.2016.0407.): LMA data as GLM proxy data that maintain the flash rate of the LMA data -> found a POD of 69%, FAR of 63 %
- Miller et al. (2015, doi: 10.15191/nwajom.2015.0308): ENTLN ground-based LJs -> CSI of about 0.1 (POD > 0.8, FAR > 0.85)
- Murphy (2017, In Eighth Conf. on the Meteorological Application of Lightning Data): NLDN ground-based LJs -> CSI of about 0.1 (POD of 0.5-0.7, FAR > 0.8)
- Farnell et al. (2017, doi: 10.1016/j.atmosres.2016.08.021): counted the thunderstorms, LJ of predictor of severe weather -> found a CSI of 0.58
- Wapler et al. (2017, doi: 10.1016/j.atmosres.2017.04.009): hail-producing thunderstorms had moderate LJs in about 70% of the cases (FAR not calculated)
- Nisi et al. (2020, doi:10.1002/qj.3897): POD for LJ-based hail warning of 0.45, FAR of 0.3
- Tian et al. (2022, doi: 10.1016/j.atmosres.2022.106404): Chinese lightning network, 2-sigma LJ algorithm to predict hail events -> CSI of 0.41 (POD of 1.0, FAR of 0.59)
And with the skill of other approaches to nowcast severe weather and rare events:
- James et al. (2018, doi: 10.1175/WAF-D-18-0038.1): operational and future thunderstorm and severity warning tools at the German weather service, POD's of 0.4-0.9 (depending the lead time) and FAR of about 0.7
- Yao et al. (2022, doi: 10.1109/JSTARS.2022.3203398): deep learning based nowcasting of heavy precipitation, CSI about 0.4-0.45 (PODs of about 0.8, FAR of about 0.6)

But then the authors go on a fishing expedition to look for connections between lightning jumps/dives and many other cloud parameters, weather parameters, with no real goal of how to use these relationships. I remind the authors that the journal is about natural hazards and hence the extensive analysis NOT related to severe weather may be better suited for another journal. I would suggest reducing the number of parameters analysed, and to focus on a key research question. Can lightning from GEO be used to supply early warnings of severe weather. Yes or no.

NHESS is a journal dedicated to research on natural hazards, and lightning itself is a natural hazard per definition. The research question you mentioned, 'can lightning from GEO be used to supply early warnings of severe weather,' has been thoroughly addressed in Erdmann and Poelman (2023). As such, we believe it is unnecessary to revisit this topic in our current manuscript focusing on thunderstorm characteristics.

The authors use qualitative descriptions of their results, while for scientific evidence we need quantitative numbers. For example, "overall similar characteristics" (line 8), "can be useful for nowcasting" (line 190), "found to resemble" (line 320), "potentially the most dangerous" (line 324), "resemble" (line 342, 346)

The authors will review the manuscript and reduce those qualitative descriptions. Please note that all quantitative values can be found in the figures and the new Table 4.

Table 3 shows that most severe weather occurs WITHOUT LJ, while most LJ would give a false alarm if used for warnings of severe weather.

Please see the papers cited above. The studies demonstrate that LJs from ground-based lightning networks give similar or higher FAR when nowcasting severe weather events than observed for the GLM LJ in Erdmann and Poelman (2023). FARs of this work are common and similar to FARs found in other studies on nowcasting of severe weather. Severe weather in general is difficult to predict, and a combination of nowcasting tools is needed as stated in the current manuscript.

Hence, there is no benefit (and maybe even a danger) in using LJ/LD in nowcasting for severe weather warnings. Why not state this?

The nowcasting performance is not the objective of this manuscript and has been addressed thoroughly in Erdmann and Poelman (2023).

Figure 1 (you mean 2?): How can we compare parameters when the normalization hides the actual values?

The normalized values allow for a conclusion which one is higher or lower. In addition, we provide values with physical units in the text and new Table 4.

The authors added a new table to the revised manuscript that displays the mean value for each characteristic and thunderstorm category. We also removed the normalized mean/median values from Fig 2 as they are not discussed in the text and complicate the readability of the figure. Also, the panel showing LDs is removed and LDs are just briefly discussed in Sect. 3.2 as we noticed that the LJs are more useful and the LD detection algorithm needs further tuning.

Line 203 states the thunderstorms with LJs cover larger areas than storms with LDs. How can we see this from normalized areas? The min and max values will be different for each subset of data.

No, the min and max are identical for each subset as the overall min and max of the entire dataset are used for the normalization.

The manuscript states (lines 152-153): "The minimum and maximum values for each characteristic are taken from all analyzed thunderstorms and do not depend on the TS category" and "X" in Eq. 1 is given as the values from all categories, hence, the entire dataset.

Most of the analysis of the cloud parameters is not new, although the lightning data set is new. We know all the connections between severe weather and cloud top, ice, overshooting tops, updrafts, etc. Why another paper on this?

The paper demonstrates that the expected results are obtained for severe thunderstorms, confirming the effectiveness of the applied methods. Additionally, in a new and interesting finding, the study reveals that thunderstorms with GLM LJs (and, to some extent, LDs) exhibit characteristics typically associated with severe thunderstorms.

In conclusion, I think this paper is very weak in results, and needs either major revision with a new focus, or should be referred to another more appropriate journal.
We would like to point out again that no one has previously studied the satellite cloud characteristics of storms with GEO-detected LJs. Our work shows that GLM LJs occur in thunderstorms with the potential to produce severe weather, although not exclusively, and not all of these thunderstorms were reported as severe.

Dear Reviewers,

Please find below my comments/suggestions regarding the manuscript entitled "Characterizing lightning jump and dive producing thunderstorms from geostationary observations".
We would like to thank the reviewer for his detailed review and constructive criticism.
Please find the authors' responses to the comments highlighted in green.

In general, I think that the Authors try to answer to many questions but I can't see a previous work for adjusting the algorithm to the requierements of the new tool, considering a similar configuration than the used in terrestrial lightning detection networks.
We would like to point out that the lightning jump algorithm in use was adapted to the GEO lightning sensor and is described in detail in Erdmann and Poelman (2023). It does not use the same configuration as the LJ algorithms for terrestrial lightning location systems (LLSs).
The commonly used algorithms was developed for lightning mapping array (LMA) with radar-based storm tracking. Erdmann and Poelman (2023) tuned the algorithm parameters to be used with GLM lightning observations and satellite-based storm cell tracking.

My specific comments are (those starting with "**" are the more rellevant ones):

- L9: "consequently leading to more structured updrafts." -> "As a consequence of more structured and intense updrafts."
OK
- L11: "GEO LJs throughout their lifecycle exhibit the most and strongest OTs, signifying highly organized updrafts, extremely cold cloud tops, and highest CRRs." Add some values.
We revised the abstract and full manuscript to provide more quantitative values. We also added the new Table 4 as a quantitative comparison of thunderstorm categories. For example, the lines above are replaced by:
"While non-severe thunderstorms have a mean cloud top temperature of 236 K, cloud tops are about 20 K colder for severe storms as well as those producing LJs and LDs. Overshooting tops (OTs) in storms producing LJs, LDs and in severe storms were about 3.4 K, 1.9 K, and 2.6 K colder, respectively, than the cloud cell as a consequence of structured and intense updrafts. On the other hand, OTs are rare and shallow in the non-severe, and thunderstorms without LJs and LDs. Accordingly, the convective rain rates (CRRs) of the LJ (23 mm/h), LD (20 mm/h) producing storms and severe storms (20 mm/h) are on average more than 3 times higher than in non-severe thunderstorms and storms without LJs or LDs. Thunderstorms experiencing multiple GEO LJs during their lifecycle feature average cloud top temperatures of 213 K, with an average of 0.5 OTs being 4.8 K colder than the anvil, and the CRR mean exceeds 26.4 mm/h."
- L17: "lightning observations can be used to locate these deep convective systems." Add some references
We added one reference (Avila et al. 2010) that cites other papers studying the relationship between lightning and convective clouds.
- L20: "lifecycle of the storm." -> Add references
We added 3 references that analyze variability of lightning activity in convective systems: Hayden et al. (2021), Borque et al. (2020), Goodman and MacGorman (1986)
Please note that all following references about LJs also show the variable flash rates in thunderstorms.
- L22: "Previous studies" -> There are many more updated refs. Please, include some of them
We removed the sentence. References are provided in the following and do not need to be repeated here.
- L25: "significant hail, or severe wind." -> Add the specific definition of each phenomenon
We added the specific values given by the NWS:

"The National Weather Service (NWS) defines severe weather as conditions involving tornadoes, significant hail (with a diameter of at least 2.54 cm or 1 inch), or winds of at least 93 km/h."

- L25: Add Farnell et al 2017

OK

- L31: "decrease in lightning activity" Some interesting references:

Murphy, M. J., & Demetriades, N. W. (2005, January). An analysis of lightning holes in a DFW supercell storm using total lightning and radar information. In Extended Abstracts, Conf. on Meteorological Applications of Lightning Data (p. 52).

We decided that we won't cite the reference as Murphy and Demetriades (2005) analyze lightning holes on a spatial scale 1-2km. Our manuscript considers the storm scale. In addition, GLM lightning data have a spatial resolution of 8-12km and, thus, features like the detailed lightning holes in Murphy and Demetriades cannot be seen at all.

Pineda, N., Rigo, T., Montanyà, J., & van der Velde, O. A. (2016). Charge structure analysis of a severe hailstorm with predominantly positive cloud-to-ground lightning. Atmospheric Research, 178, 31-44.

Pineda et al. (2016) analyze one specific storm in Catalonia, that showed a slight decrease in the total flash rate prior to hail reports twice. We added it as reference.

- L38: "appear to be" -> "are"

OK

- L39: "CG records" -> Add Rigo, T., & Farnell, C. (2022). Characterisation of thunderstorms with multiple lightning jumps. Atmosphere, 13(2), 171.

OK

- L53: "severe weather" -> Add Farnell et al (2017)

We believe the provided reference does not fit the context of the sentence as Farnell et al. (2017) did not optimize the LJ algorithm but just applied it.

- L66: "found" -> add Farnell et al 2018

We acknowledge the Referee's significant contributions to the studies suggested for citation (e.g., Rigo and Farnell 2022; Farnell et al. 2018, 2017; Pineda et al. 2016; Rigo and Llasat 2016; Rigo et al. 2010). While we have now cited several of these works, particularly the more recent ones, we believe that it is not necessary to include all of them for the purposes of our manuscript..

- General: add a list of acronyms

All acronyms are defined upon their first use, in accordance with the journal's guidelines.

- L151: "characteristics are normalized" -> "similar to Rigo and Llasat (2016) or Rigo et al (2010)"

Min/max normalization is a widely used approach and is not unique to the suggested reviewer's papers, which even employ a different normalization method. Therefore, we do not believe it would be beneficial to cite these papers.

- L164: "increase level (RIL) algorithm." -> add references

The algorithm type was developed in Erdmann and Poelman (2023), and the reference is given in the previous sentence.

We updated the LJ algorithm description so that the revised manuscript does not mention all the different algorithm types. We are simply providing the name and short description of the FRarea LJ algorithm that was developed and optimized for the data used in the manuscript. Hence, no further adaption was needed.

** L162: "Erdmann and Poelman (2023) optimized the LJ algorithm for GLM lightning records" -> Have hou considered the limitations of GLM? Have you compared the FR with the obtained by the NLDN or a LMA, for example?

Yes, Erdmann and Poelman (2023) considers GLM as a new tool, taking into account the strengths and weaknesses of optical lightning detection from space. Comparing the FR to ground-based systems would not provide added value for the LJ detection (The comparison of GLM to terrestrial LLSs is provided in much detail by other authors [e.g., Marchand et al. 2019; Zhang and Cummins 2020, Murphy and Said 2020, Rutledge et al. 2020].). Erdmann and Poelman (2023) tested a variety of different FR thresholds and picks the most suitable one for the GLM data, and the same FR

threshold is used in this submitted manuscript. In addition, also the most suitable sigma threshold is identified for the GLM lightning records in combination with satellite-based cell tracking.

We are aware that GLM performs less effectively over the northern CONUS compared to the southeastern CONUS, and that situations involving high flash rates, short durations, and small flashes present challenges for optical lightning detection from space. The appendix of the submitted manuscript addresses the issues of spatial variation in the GLM detection efficiency and shows that there is small impact on the LJ detection.

** L166: "The FRarea LJ algorithm first smoothens" Why? In the case of large thunderstorms, you limit the algorithm capabilities

The FRarea algorithm is defined like that as it is a modification of the sigma LJ algorithm that smoothens the time series to 2-minute steps (Gatlin and Goodman 2010, Schultz et al. 2009). The normalization per cell area does not bring any drawback since the calculation uses the ratio. If a cell maintains its surface area, then the areas simply cancel out and the algorithm is identical to the sigma LJ algorithm. In addition, the FRarea algorithm better accounts for the merging/splitting of cells that change size rapidly from one time step to the next. Erdmann and Poelman (2023) show that this normalization increases the skill slightly compared to the original sigma LJ algorithm.

** L171: "This study uses the FRarea LJ algorithm with FR threshold of 15" This threshold is for the observed FR or the normalized FR?

The FR threshold is always based on the observed flash rate. Only the sigma calculation takes the normalization into account (since the history is important for sigma only). We added the unit in the paper, so it is clear that the FR threshold is not normalized:

"The LJ algorithm used in this study is the FRarea LJ algorithm, optimized for GLM lightning records as detailed by Erdmann and Poelman (2023). With a FR threshold of 15 flashes per minute and a sigma level of 1.0, the algorithm first checks …"

** L186: "an LJ also had an LD" -> Why?

We assume that most storms with a sufficiently high flash rate to trigger the LD algorithm feature a reduction of the flash rate (FR) during the decaying phase, and that this reduction often creates LDs. Since the FR threshold of the LD algorithm was set lower than for the LJ algorithm (after analyzing CSI values), LDs are detected for all the LJ storms. However, LDs have never been studied, and further research would be needed to understand a possible correlation between LJs and LDs. Please note that these lines were removed and LDs are now discussed in the Sect. 3.2.

- L187: "However, it is possible that these storms did produce severe weather that was not reported." There are many references about the limitations of severe weather databases.

Hulton, F., & Schultz, D. M. (2024). Climatology of large hail in Europe: characteristics of the European Severe Weather Database. Natural Hazards and Earth System Sciences, 24(4), 1079-1098. Schroeter, S., Richter, H., Arthur, C., Wilke, D., Dunford, M., Wehner, M., & Ebert, E. (2021). Forecasting the impacts of severe weather. Australian Journal of Emergency Management, The, 36(1), 76-83.

Thank you, we added the references to the manuscript.

"However, it is possible that these storms did produce severe weather that was not reported since severe weather databases have documented limitations (e.g., Hulton and Schultz 2024, Schroeter et al. 2021)."

** L195: "The key questions to be answered are (i) Do thunderstorms with GLM LJs and/or LDs feature particular characteristics?, (ii) How do the severe thunderstorms compare to the thunderstorms with GLM LJs and/or LDs?, (iii) Do GLM LDs provide added value?, and (iv) Is the number of GLM LJs or LDs important?" These lines should be moved to the end of the Introduction, as the goals of your research

OK, we moved them to the introduction. We also modified the research questions to better match the manuscript content and objective:

"(i) What do GLM LJs tell us about the storms structure from a satellite point of view?, (ii) Are GLM LJs useful to assess thunderstorm severity?, (iii) Do GLM LDs provide additional information about the thunderstorms?"

Sect. 4 is adapted accordingly.

- L201: "Figure 2 compares the normalized characteristics" Why this comparison?

We want to identify cloud characteristics of storm with LJs (and LDs), and we use the thunderstorms without LJs as a reference to know what distinguishes a thunderstorm with a GLM LJ from a thunderstorm that did not produce a LJ. This is among the main objectives of this paper.

** L203: "LJs (Figure 2b) cover a larger area than the storms with LDs" But all cells with LJ have LD... I can't understand this result

The storms with LJs are a subset of the storms with LDs. Hence, there are several storms that have an LD but no LJ, and these storms cause the difference.

- L204: "((Figure 2a))." Typo

Corrected

** L205: "large anvils for CTs near the tropopause" -> Explain better

Satellite-based cell detection and tracking sees the cloud tops. If a thunderstorm reaches the tropopause and an anvil forms, this cell appears larger than a thunderstorm that features mainly vertical development and has no anvil.

We modified the sentence in the manuscript:

"This could be related to the formation of large anvils for CTs near the tropopause that acts as a natural ceiling. If a thunderstorm grows up to the tropopause, vertical development is hampered and moist air is forced horizontally. The satellite-based cell detection sees the resulting anvil and those cells appear larger than thunderstorms that grow mainly vertically."

** L206: "Storms with LJs had an average area of 15,780 km2 (median 6,995 km2), whereas non-severe thunderstorms and those without LJs typically covered about 2,000 km2 on average (with medians around km2)." -> Considering that most of supercells are associated with severe weather (Farnell, C., Rigo, T., & Pineda, N. (2018). Exploring radar and lightning variables associated with the Lightning Jump. Can we predict the size of the hail?. Atmospheric Research, 202, 175-186.), do you have an approximation of the area of this convective mode in the region of study?

We use only satellite data and did not separate the thunderstorms by type/ convective mode. That means we do not know which thunderstorms are supercells (single cells, multi cells, or embedded in MCSs). An idea of the overall area of the storms is provided in Table 2 of Erdmann and Poelman (2023) that uses basically the same dataset as this manuscript:

Mean (median) area of thunderstorms: 2988.5km2 (762.8km2)

Mean (median) area of of all cloud cells: 619.9km2 (146.6km2)

-> the LJ storms are exceptionally large in their footprint area.

- L209: "LJs, LDs, and/or NCEI reports." Could it be related with the way as you consider the FR?

We don't think the FR has an impact on the cell area. It's possible that some cells exist with extensive cloud shields that have low flash rates and do not produce severe weather.

** Section 3.1.1: You could compare your results with:

Bedka, K. M. (2011). Overshooting cloud top detections using MSG SEVIRI Infrared brightness temperatures and their relationship to severe weather over Europe. Atmospheric Research, 99(2), 175-189.

Bedka, K., Brunner, J., Feltz, W., & Dworak, R. Overshooting Top and Enhanced-V Detections Objective Day/Night Overshooting Top and Enhanced-V Detections Using MODIS, AVHRR, and MSG Imagery in Preparation for GOES-R ABI.

The references given above mention a correlation between OTs and severe weather. We added another reference of this author in the following sentence as a reference for typical BTs of cloud tops:

"Coldest CT temperature is found for the multiLJ TSs (mean of 213 K). Typical BT thresholds for the detection of overshooting tops are in the range of 215 K and high anvils with 225 K (Autones et al, 2020, Bedka et al, 2016). Hence, CTs of the multiLJ storms are extremely cold."

** L239 "Overshooting tops (OTs) define a region of" How do you detect OTs?

This is part of the NWCSAF software. We added the following description to the manuscript, in a new paragraph below the section describing the NWCSAF software:

"OTs are detected in the NWCSAF RDT package through the application of several temperature and BTD criteria. The software identifies extremely cold cloud pixels (colder than -223 K in the mid-latitudes), and compares them to the surrounding pixels to identify the depth (as the temperature difference, DT) and horizontal extent of the OT. The BTDs take channels of WV6.2, WV7.3, and IR10.8 into account. Satellite pixels can also be identified as OT if they are at least 5 K colder than the tropopause. A detailed description of the OT detection algorithm can be found in Autones et al. (2020, p. 49-50)."

** L240: "above an anvil" -> Include at least one reference: Bedka, K. M., & Khlopenkov, K. (2016). A probabilistic multispectral pattern recognition method for detection of overshooting cloud tops using passive satellite imager observations. Journal of Applied Meteorology and Climatology, 55(9), 1983-2005.

We moved this description of OTs to the methods and add the reference there.

- L240: "Sometimes OTs even break through the tropopause. OTs are usual transient features, so this study analyzes the maximum OT activity of each thunderstorm. OT development needs a strong force manifestated as a strong, persistent updraft in thunderstorms. The air gets accelerated vertically and can overshoot the level of thermal equilibrium. Hence, OTs are indicative of dynamical thunderstorm cells with strong updrafts that are usually well organized. Given that strong updrafts frequently play a crucial role in the formation of tornadoes and large hail, storms with these characteristics are especially significant for nowcasting." These lines should be removed from results.

OK. We moved these lines to the methods and new paragraph about detection of OTs, and shortened this general part.

- Section 3.1.2: Include at least one reference: Bedka, K. M., & Khlopenkov, K. (2016). A probabilistic multispectral pattern recognition method for detection of overshooting cloud tops using passive satellite imager observations. Journal of Applied Meteorology and Climatology, 55(9), 1983-2005.

We moved this description of OTs to the methods section and add the reference there.

** Section 3.1.3.: "The max CRR in Figure 2 reveals that thunderstorms with LJs experience higher rain rates than storms with LDs, while CRRs 260 of the latter are still significantly higher compared to the noLJ storms. Furthermore," Explain how the CRR are estimated or reference it

This is also part of the NWCSAF nowcasting software. We added the reference to the documentation in the manuscript, under the method section describing the NWCSAF software:
"The NWCSAF software also includes a dedicated package to estimate convective rain rates (CRRs). This estimation uses analytic function calibrated based on radar data as ground truth, and also takes lightning observations into account. The complex algorithm to estimate CRRs is detailed in Lahuerta et al. (2020, p. 22-41)."

- L293: "but about half of the LD storms had no LJ. " Why?

The number of LDs storms was much higher than the number of LJ storms. This is a finding based on the applied detection algorithms that were optimized to match severe weather reports. Different thresholds in the algorithms would change this ratio.

Other questions:
- Which is the Lead time between LJ and severe weather?

Please see Erdmann and Poelman (2023, section 4c). Leadtimes are very variable, with mean (median) leadtimes of 37.5 min (34.0 min).

We added the information in the introduction:
"Erdmann and Poelman 2023 were among the first to optimize the LJ detection specifically for GLM lightning records in the central and eastern contiguous United States (CONUS) and found that GLM LJs as severe weather predictors reach a critical success index of about 0.5, with leadtimes averaging about half an hour."

- which is the Lead Time between LD and severe weather?

We have not analyzed the leadtime for LDs yet as this work shows that the LD algorithm should be further modified and also that LJs seems to be more meaningful than the LDs.

- Have you considered the limitations of the occurence area? Problems in the limits of the region, or in higher latitudes?

Please see the paper appendix.

- Have you considered the size of the thunderstorms? In some cases it is possible that thunderstorm have small dimensions, specially in non warm season.

Why is the size of the storms important? We understand that it will be meaningful for the overall flash rate. If we look for LJs, however, the relative rate of change for one storm cell is interest. It should not matter too much whether the detected cell is initially small or large.

Best regards

---

## Author Response (AR3)

Dear Authors,
Thank you for your hard work. I have some questions that you should address before accepting the manuscript. Please, find them in the attached doc.
All the best.

Thank you very much for reviewing our manuscript again.
Please note the authors' comments on the review highlighted in green.

Attached file with this report:

Dear Authors,
Thank you very much for your detailed answers and the notable improvement of your manuscript. The new version shows a better organization of your Research and provide more curated information about your findings.
Please find below my new comments, some of them appeared from your previous answers (which are marked in bold and between quotation marks):

**"We would like to point out that the lightning jump algorithm in use was adapted to the GEO lightning sensor and is described in detail in Erdmann and Poelman (2023). It does not use the same configuration as the LJ algorithms for terrestrial lightning location systems (LLSs). The commonly used algorithms was developed for lightning mapping array (LMA) with radar-based storm tracking. Erdmann and Poelman (2023) tuned the algorithm parameters to be used with GLM lightning observations and satellite-based storm cell tracking."**
I fully understand your comment, but after carefully reading the two manuscripts (the presented one and the 2023 one), I think that you should include a graphic which could help the reader understand the differences between the original and the proposed version of the algorithm. This scheme should present the life cycle of a thunderstorm from the point of view of the lightning activity and how you transform in the sigma variable and when a jump or a dive occurs.

Please see below a time series showing one thunderstorm trajectory with LJs identified using the original 2σ and our adapted FRarea LJ algorithm (Figure R1). The figure also includes the flash rate and algorithm flash rate (FR, blue) and σ-level thresholds (grey), as dashed line for the original 2σ LJ algorithm, and as solid lines for our FRarea LJ algorithm. Detected LJs are shown as red markers as indicated in the legend. The time evolution of the cell area with 10-minute updates is plotted in the 2 row of Figure R1.
The FRarea LJ algorithm of this study considers the cell area when calculating σ-levels. Hence, the solid and dashed lines for the σ-levels of our and the 2σ LJ algorithm, respectively, differ especially at times when the cell area is changing. This behavior is evident during the start of the trajectory, where the cell grows, and more flashes can be matched to the larger footprint. The original 2σ LJ algorithm detected a LJ, whereas our LJ algorithm didn't as the FR per area did not increase sufficiently and σ(area) remained close to 0. Note also the decreasing σ(area)-level for the increase in FR at about 40 minutes, happening with a simultaneous jump in the cell area likely due to merging of cells. The raw σ-level increased at this time. The opposite behavior can be seen towards the end of the life cycle (at about 94 min): The σ(area)-level exceeds the raw σ-level as the FR increased within a shrinking, soon decaying cell. At 36 min, only

our FRarea LJ algorithm could detect the LJ that actually led to maximum FR of 93 flashes/min. The lower σ-level threshold of our FRarea algorithm of 1 compared to the original 2σ algorithm helped to detect this LJ that should not be missed. After 64 min of the cell lifetime, both algorithms detected a LJ.

[Figure]

Figure R1: LJ algorithm comparison for a thunderstorm trajectory on 10 Jun. 2020

We decided that we won't use this exact Figure in the manuscript. The 2-sigma algorithm is not explained and not part of this work so it would rather confuse the reader. Instead, we add a figure showing the time series with LJs and LDs and the parameters as obtained from the applied algorithms to identify LJs and LDs. The following paragraph and figure were added to Section 2.6:
""

Figure 2 illustrates the application of the LJ and LD detection algorithms for one thunderstorm trajectory starting on 06 Feb. 2020 at 0520 UTC and lasting almost 90 minutes. The thunderstorm reached a maximum FR of 48 flashes per minute about 75 minutes after the cell had been identified, with a second FR peak observed 54 minutes after the start. In total, 2 LJs and 3 LDs were detected as indicated by the red markers. The detection algorithm thresholds are also shown as horizontal lines, in blue for the FR threshold and in grey for the σ-level threshold. The flash rate must be greater than the FR threshold to detect a LJ or LD. At the same time, the σ-level should exceed the threshold for the LJ algorithm, and be more negative than the threshold for the LD algorithm. The σ-level peaked during the first LJ. Although the raw FR increased more rapidly during the second than during the first LJ, the simultaneous growth of the cell led to a smaller σ-level than in the first LJ, as the LJ algorithm accounts for cell area by dividing FR by the cloud cell area.

[Figure]

Figure 2. Flash rate (blue shading), σ-level (black) and detected LJs and LDs (red triangles) for one thunderstorm trajectory starting 06 Feb. 2020, 0520 UTC. The FR-thresholds (blue) and σ-level thresholds (grey) of the LJ (solid) and LD (dashed) detection algorithms are shown as horizontal lines (dashed lines superimposed).
"""

**"Why is the size of the storms important?"**
The difference in size between supercells or normal cells is quite evident and is an indicator of the intensity and wide of the updraft. This is only one example, but there are many more. For example, large squall lines or MCS tend to last more and to produce more severe weather reports, because the organization is higher and helps to extend the duration. These are two examples of the importance of the size. For sure, there are cases where the size does not provide any relevant information, but these examples are less associated with convection.
In fact, you say in your manuscript that: "the mean cell areas in Table 4 confirm the previous finding. On average, severe thunderstorm cells covered an area of 12,812 km2 (median 4,089 km2). Storms with LJs had an average area of 15,780 km2 (median 6,995 km2), whereas while non-severe thunderstorms and those without LJs typically covered about 2,000 km2 on average, with medians around 550 km2)."
We agree with the referee that different storm types such as supercells, single cell storms, or MCSs are known to have different sizes. We found that the thunderstorms with LJ and the severe storms feature larger footprint areas than the no-LJ and non-severe storms. Hence, this could mean that more organized storm types prevail in these 2 categories.
We added this aspect to the discussions (section 3.1.1):
"The fact that thunderstorms with LJs, LDs and the severe storms covered larger areas than the average area of all thunderstorms may indicate an above-average fraction of well-organized thunderstorm types like supercells, multi-cell storms, or MCSs that are known for larger footprint areas than the ordinary single cell thunderstorms."

However, we did not analyze or separate different storm types. The study analyzes almost 25k storm trajectories, and would take a lot of time to evaluate those trajectories by hand for the thunderstorm type. We may look into this aspect in the future for certain selected trajectories.

About the list of acronyms, I know that is not mandatory, but in your case can result very helpful to the reader, because the large number of items. It is your choice.
We didn't add a list since we think it is uncommon in journal articles.

Some new comments:
- Typo in caption of table 4: "Torndao" should be "Tornado"
OK
- I suggest merging subsections 3.1.2 and 3.1.3, reducing part of the text. There a lot of results and I suggest you that focus on the main items.
We merged the sections and shortened parts of the text.
- You say that "It should be mentioned that severe weather is observed in storms without LJs, and that there are non-severe storms that had GLM LJs". Fortunately, there is not an algorithm able to reproduce the exact behavior of a thunderstorm. Furthermore, as you also indicate in the manuscript (and we previously discussed), there are limitations on the direct ground observations of severe weather. Because of this, I suggest that you replace "Hence, the GLM LJs should not be used as standalone severe weather warning tool but in combination with other data." By "We recommend the algorithm users to consider its limitations, in special at the time of applying for operational purposes. The algorithm can indicate the occurrence of severe weather in a thunderstorm but, as occurs with all the real-time tools, the user must consider many other elements, such as signatures observed in radar imagery, satellite data or terrestrial lightning detection networks."
Thank you for the suggestion, we changed the text accordingly and added a slightly modified sentence to the manuscript: "Users of the algorithm are advised to be aware of its limitations, especially when using it for operational purposes. The algorithm can indicate the occurrence of severe weather in a thunderstorm, but as with all real-time tools, the user must take into account many other elements, such as signatures observed in radar images, satellite data or terrestrial lightning detection networks."

The revised paper is much improved. I think the paper can be published after additional minor corrections.

We thank the referee for his effort to review our manuscript and to help improve it. We appreciate the positive feedback. The new comments will be included in the text as stated in our answers to the comments highlighted in green.

Line 5: and/or severe weather

OK

Line 9: in the non-severe storms

OK

Line 13: Is 26.4 mm/h significantly different from 20 mm/hr?

These are average values. We can have a look at Figure 3b [new: Figure 4b]: The IQR of the withNCEI storm as severe storms, with average CRR of 19.5 mm/h (rounded to 20 mm/h in the abstract), ranges from 9 to 28 mm/h. Hence, it includes the 26.4 mm/h value. CRRs show a wide range of values and the category IQRs approximate 20 mm/h. We added this aspect to the discussion (Section 3.1.4 [new: 3.1.3]): "However, one needs to consider the high variability of CRRs in each TS category expressed as IQRs of about 20 mm/h.".

We are convinced that 26.4 mm/h means a significant increase to 19.5 mm/h as (i) it's a relative increase of about 35% (=(26.4-19.5)/19.5), and (ii) 26.4 mm/h is almost as high as the 75% of the severe storms' distribution (28 mm/h).

The impact of 6.9 mm/h rainfall for events like flash floods depends on additional criteria like storm moving speed, the surface characteristics, and terrain.

Line 27: In addition

OK

Line 80: data are ingested

OK

Line 166: Why these specific 14 parameters? Are these all the parameters provided, or did you choose a subset?

As you may see in the peer review process. initially, we used all parameters that our nowcasting software provides us. We reduced the number of parameters in the paper to improve the readability. These remaining 14 parameters mean a trade-off between as much information as possible about the storm cells and the most important findings.

Line 236: How do you measure pressure? Is this an independent parameter, or derived from the brightness temperature? Table 4 should have independently observed or calculated parameters.

The NWCSAF nowcasting software takes ECMWF NWP as an additional input. Pressure measures are based on the NWP data taking the brightness temperatures and NWP temperature profiles into account. The NWCSAF software provides pressure as an output.

Line 257: Where do we see these results for withHail and withTornado?

We decided that we would not show the results for the individual severe weather types (they were included in a previous submission of the manuscript), and neither to discuss them in detail.

We think it is worth mentioning the results here as they differ from all severe storms. We added "(not shown)" to the text so that it is clear that this is an additional information.

Line 305: "colder" and "lower" than what?

We added that storms with LDs are the reference for the comparison.

Line 336: optical storms?

A mistake resulting from the last revision. We corrected the sentence: "For the first time, thunderstorms with GEO LJs and/or LDs detected from optical lightning observations are characterized in detail."